# Revisiting Dynamic Graph Clustering via Matrix Factorization

## Abstract

Dynamic graph clustering aims to detect and track time-varying clusters in dynamic graphs, revealing the evolutionary mechanisms of complex real-world dynamic systems. Matrix factorization-based methods are promising approaches for this task; however, these methods often struggle with scalability and can be time-consuming when applied to large-scale dynamic graphs. Moreover, they tend to lack robustness and are vulnerable to real-world noisy data. To address these issues, we make three key contributions. First, to improve scalability, we propose temporal separated matrix factorization, where a single matrix is divided into multiple smaller matrices for independent factorization, resulting in faster computation. Second, to improve robustness, we introduce bi-clustering regularization, which jointly optimizes graph embedding and clustering, thereby filtering out noisy features from the graph embeddings. Third, to further enhance effectiveness and efficiency, we propose selective embedding updating, where we update only the embeddings of dynamic nodes while the embeddings of static nodes are fixed among different timestamps. Experimental results on six synthetic and five real-world benchmarks demonstrate the scalability, robustness and effectiveness of our proposed method.[1]

## CCS Concepts

• **Computing methodologies → Factorization methods**.

## Keywords

Graph Clustering, Temporal Networks, Community Detection.

**ACM Reference Format:**
Anonymous Author(s). 2018. Revisiting Dynamic Graph Clustering via Matrix Factorization. In *Proceedings of Make sure to enter the correct conference title from your rights confirmation emai (Conference acronym 'XX)*. ACM, New York, NY, USA, 18 pages. https://doi.org/XXXXXXX.XXXXXXX

## 1 Introduction

Dynamic graph clustering, also known as dynamic community detection, aims to leverage graph topological structures and temporal dependencies to detect and track evolving communities [6, 37, 51]. As an effective tool to reveal the complex evolutionary rules behind complex real-world systems, dynamic graph clustering has drawn great attention in various fields, such as social analysis [20, 33, 79, 84], recommendation [59, 67, 76, 80], and AI4Science [21, 57, 60].

---

[1]For reproducibility, our code will be available on Github with the following anonymous version https://anonymous.4open.science/r/DyG-MF

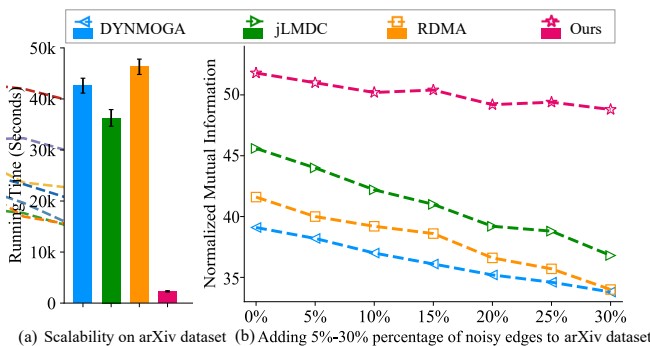

(a) Scalability on arXiv dataset    (b) Adding 5%-30% percentage of noisy edges to arXiv dataset

**Figure 1: Running time and performance on noisy data of our proposed method and three matrix factorization-based methods, *i.e.*, DYNMOGA [15], jLMDC [30], and RDMA [51].**

Much research in recent years has been proposed for dynamic graph clustering, which can be broadly classified into two classes [53]. *(i) Neural Network-based methods* [16, 38, 69, 71] generally focus on learning dynamic node embedding, with clustering methods often applied as a post-processing step. These methods separate node embedding learning and clustering into two independent steps, leading to sub-optimal performance [12, 30]. To relieve this issue, *(ii) Matrix Factorization*-based methods [5, 10, 30, 35, 36, 41] have been widely proposed. These methods can jointly optimize clustering and dynamic node embedding learning simultaneously, *i.e.*, they cluster nodes at each timestamp while maintaining temporal smoothness of node embedding among different timestamps, achieving overall optimal performance on many benchmarks.

Despite the great success of matrix factorization-based methods, there are still two challenges. *(i) Weak Scalability.* Matrix factorization is an NP-hard problem with a time complexity of approximately $O(n^3)$ and a space complexity of $O(n^2)$ for a single graph containing $n$ nodes [44, 64]. A pre-experiment is shown in Figure 1(a), best-performing matrix factorization-based baselines require approximately 40,000 seconds to process the arXiv dataset, which contains about 30,000 nodes, limiting their applicability to real-world dynamic graphs with millions of nodes [29]. *(ii) Low Robustness.* Real-world dynamic graphs contain noise and missing data, which disrupt their regular evolution patterns and pose significant challenges for dynamic graph clustering [73, 78]. A case study is shown in Figure 1(b), adding random noisy edges to dynamic graphs leads to a sharp performance drop of these baselines.

To address these issues, we propose a scalable and robust dynamic graph clustering framework via seperated matrix factorization, called *DyG-MF*, containing three key contributions. Firstly, to enhance scalability, we propose *(i) Temporal Separated Matrix Factorization*. We apply temporal matrix factorization in a "divide and conquer" manner [34, 58], where we randomly divide the nodes into subsets and transform the original large-scale matrix factorization problem into several independent matrix factorization of these subsets. Since matrix factorization is applied separately to these

smaller subsets, it also reduces computational cost. To achieve this, we design the temporal landmark selection, ensuring coherence of node embeddings across different subsets at current timestamp and maintaining consistency of node embedding between different timestamps. Secondly, to improve robustness, we introduce *(ii) Bi-clustering Regularization*, which reduces the impact of noisy features on dynamic graph clustering by optimizing the rank of the matrix. We further proof this regularization can be spreadable and applied as a constraint in the matrix factorization of each node subset. Finally, to further enhance effectiveness and efficiency, we propose *(iii) Selective Embedding Updating*. We first divide the nodes into dynamic and static groups by jointly considering their topological and embedding changes. We then only update node embeddings of the dynamic group while keeping the node embeddings in the static group fixed across different timestamps. The main contributions of this study can be summarized as follows.

- To enhance scalability and efficiency of matrix factorization-based methods, we design a temporal separated matrix factorization framework, where we divide a single large matrix into multiple smaller matrices for independent factorization.
- To improve robustness, we propose separable bi-clustering regularization to filter out noisy features from node embeddings.
- To further enhance effectiveness and efficiency, we propose selective embedding updating, where only the node embeddings of the dynamic group are updated at each timestamp.
- Experimental results on 11 benchmarks demonstrate the scalability, robustness, efficiency, and effectiveness of DyG-MF.

## 2 Related Work

**Neural Network-based Methods.** Some neural network-based methods employ coupled approaches, which first condense dynamic graphs into one static graph and then apply clustering methods, such as CNN-based [55, 82] and GNN-based methods [74, 83], to identify clusters. Other methods employ two-stage approaches, which first learn dynamic graph embeddings [2, 18, 75] and then apply clustering methods to these embeddings to identify clusters [8, 9, 40, 48, 68, 81]. For example, RNNGCN [69] and DGCN [16] use RNNs or LSTM to capture temporal dependencies for graph embeddings, which are then clustered using graph convolutional layers. ROLAND [71] extends static GNN-based graph embedding methods to dynamic graphs by using gated recurrent units to capture temporal information. To reduce time consumption, SpikeNet [32] uses spiking neural networks to model the evolving dynamics of graph embeddings, achieving better performance with lower computational costs. For more related work, refer to [4, 28, 77, 85]. The main issue with neural network-based methods is their separation of dynamic graph embedding and clustering into two independent processes, making it difficult to ensure that graph embedding provides the most suitable features for clustering [12, 30]. Furthermore, most of them face weak scalability and interpretability issues on large-scale graphs [22, 65]. Thus, we focus on separated matrix factorization, jointly optimizing dynamic graph embedding and clustering, and improving scalability and interpretability (Fig.7(C)).

**Matrix Factorization-based Methods.** Matrix factorization-based methods cluster nodes at each timestamp using matrix factorization while optimizing the temporal smoothness of node embeddings

among different timestamps. Recently, numerous methods with different strategies have been proposed to improve temporal smoothness. For example, sE-NMF [42], jLMDC [30], and NE2NMF [31] estimate temporal smoothness by analyzing topology changes between graphs at the current and previous timestamps, while PisCES [35] smooths clusters by considering topology changes across the entire dynamic graph. In contrast, other methods use clustering metrics or reconstruction loss to measure temporal smoothness. For example, DynaMo [86] improves temporal smoothness by incrementally maximizing modularity between successive graphs, and PMOEO [56] and MODPSO [70] employ evolutionary algorithms to minimize the NMI of clusters across different timestamps. ePMCL [67] and HMM-MODCD [1] use hidden Markov models to reconstruct clusters from previous timestamps and capture evolutionary patterns. Although these methods can simultaneously optimize clustering accuracy and temporal smoothness, they often suffer from low robustness and lack fine-grained node-level temporal smoothing strategies. In this study, we address these issues and enhance robustness, scalability, and practicality for large-scale real-world dynamic graphs.

## 3 Preliminary

**Dynamic Graph Clustering.** We consider a dynamic graph as a sequence of snapshots and the $t$-th snapshot $\mathcal{G}_t = (\mathcal{V}_t, \mathcal{E}_t)$, defined for $0 \leq t \leq \tau$. Here, $\mathcal{V}_t$ and $\mathcal{E}_t$ represent the set of nodes and edges in the $t$-th snapshot. Let the graph contain $n$ nodes and $W_t \in \mathbb{R}^{n \times n}$ and $M_t \in \mathbb{R}^{n \times n}$ represent the weighted adjacency matrix and pointwise mutual information matrix [50] for the $t$-th snapshot $\mathcal{G}_t$, respectively. In $M_t$, each element $m_{ij} = \log \frac{w_{ij} \sum_k d_k}{d_i d_j}$ with $d_i$ as the degree of the $i$-th node. Dynamic graph clustering seeks to detect a set of non-overlapping clusters for $\mathcal{G}_t$, which corresponds to a partition of $\mathcal{V}_t$. This partition is represented as $\mathcal{V}_t = \{V_{i,t}\}_{i=1}^{\varrho_t}$, where $\varrho_t$ represents the number of clusters.

**Matrix Factorization.** We first introduce a matrix factorization-based baseline, which we refer to as temporal matrix factorization. Inspired by Qiu et al. [50], factorizing pointwise mutual information (PMI) matrix $M_t$ is equivalent to Skip-gram-based graph embedding [49], which can encode graph topology information. Using matrix factorization, graph embedding can be formulated as:

$$\mathcal{L}_t^{\text{Sc}} = \|M_t - C_t H_t\|_F^2, \tag{1}$$

where $\mathcal{L}_t^{\text{Sc}}$ represents the snapshot clustering (Sc) cost, each row of $C_t \in \mathbb{R}^{n \times r}$ denotes the embedding of the corresponding node, each column of $H_t \in \mathbb{R}^{r \times n}$ denotes the embedding of the corresponding node when it is considered as context for other nodes, and $r \ll n$ is the number of selected features at timestamp $t$.

Jointly considering node embedding learning and clustering can mutually reinforce each other, *e.g.,* node embedding learning can select the most suitable features for clustering. Inspired by Chris Ding et al. [11], adding non-negativity and normalization constraints to matrix factorization of Eq.(1) makes it equivalent to spectral clustering. Therefore, $\mathcal{L}_t^{\text{Sc}}$ can be re-formulated as follows:

$$\mathcal{L}_t^{\text{Sc}} = \|M_t - C_t H_t\|_F^2, \quad s.t. \ C_t \geq 0, \ H_t \geq 0, \ C_t \mathbf{1} = \mathbf{1}, \tag{2}$$

where each row of $C_t$ not only represents the embedding of the corresponding node, but also represents the clustering index of the node, *i.e.,* the rank of the maximum value in each row represents its

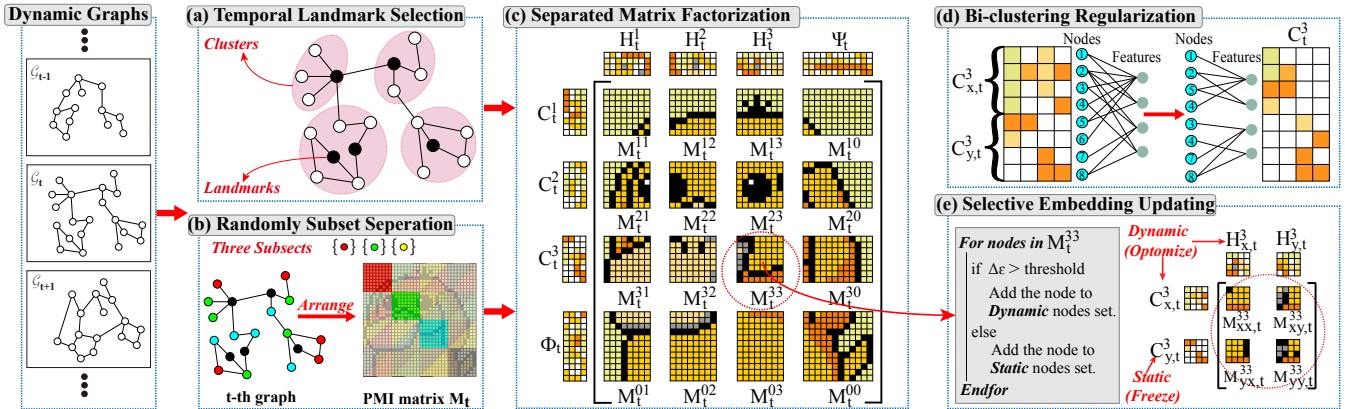

**Figure 2: Overview architecture of proposed DyG-MF. Our method (a) first selects temporal landmarks and (b) randomly divides nodes into several groups for (c) separated matrix factorization ((a)-(c) introduced in Sec 4.1). In addition, we apply (d) bi-clustering regularization (Sec 4.2) and (e) selective embedding updating (Sec 4.3) to dynamic graph clustering.**

corresponding clustering category. For example, the $i$-th node with three dimensions $C_{i,:}=[0.2, 0.7, 0.1]$ belongs to the second cluster.

Incorporating temporal information to constrain node community changes between consecutive timestamps consistently can always enhance the accuracy of graph clustering. We can simply define the temporal smoothing (Ts) cost as follows:

$$\mathcal{L}_t^{\text{Ts}} = \|C_t - C_{t-1}\|_F^2, \quad s.t. \ C_t \geq 0, \ C_t \mathbf{1} = \mathbf{1}. \tag{3}$$

Based on Eqs.(2) and (3), we obtain overall objective function as:

$$O_t = \mathcal{L}_t^{\text{Sc}} + \alpha \mathcal{L}_t^{\text{Ts}}, \quad s.t. \ C_t \geq 0, \ H_t \geq 0, \ C_t \mathbf{1} = \mathbf{1}, \tag{4}$$

where $\mathcal{L}_t^{\text{Sc}}$ measures the cluster quality of the $t$-th snapshot, $\mathcal{L}_t^{\text{Ts}}$ measures the differences of clustering results between the $t$-th and the $(t-1)$-th snapshots, and $\alpha$ is a hyperparameter to balance the importance of there two items (we set $\alpha=0$ when $t=1$). Please note that a higher $O_t$ indicates worse clustering quality or smoothness.

## 4 Methodology

As shown in Figure 2, our method DyG-MF consists of three main components: **(i) temporal separated matrix factorization** jointly learns graph embedding and clustering in Sec 4.1, **(ii) bi-clustering regularization** reduces noise and enhance robustness in Sec 4.2, and **(iii) selective embedding updating** aims at better embedding alignment in Sec 4.3. We will introduce each component in order.

### 4.1 Temporal Separated Matrix Factorization

Directly optimizing Eq.(4) is unacceptable time-consumption for large-scale dynamic graphsm, since its time and space complexity is $O(n^3)$ and $O(n^2)$ for a graph with $n$ nodes. To solve this issue, we propose temporal separated matrix factorization which transforms one large matrix factorization problem into several small matrix factorization sub-problems. The key point is to select a few nodes, called landmarks, to ensure consistency and coherence in node embeddings across all small matrix factorization.

**Temporal Landmark Selection in Fig 2(a).** A simple idea is to select the nodes closest to each cluster center as landmarks, ensuring that these nodes are representative at the current timestamps.

If we follow K-means clustering, the $l$-th cluster centers at the $t$-th timestamp $\theta_{l,t}$ can be found by repeating the following process:

$$\underset{\{\Theta_{l,t}\}_{l=1}^{\varrho_t}}{\arg\min} \sum_{l=1}^{\varrho_t} \sum_{a \in \Theta_{l,t}} (\|\boldsymbol{m}_{a,\cdot,t} - \boldsymbol{\theta}_{l,t}\|_2^2), \tag{5}$$

where $\boldsymbol{m}_{a,\cdot,t}$ is the $a$-th row vector of $M_t$, $\varrho_t$ is the number of cluster centers, automatically determined by the elbow method [63], $\{\Theta_{l,t}\}_{l=1}^{\varrho_t}$ represent current nodes' clusters, $a \in \Theta_{l,t}$ indicates that the $a$-th row vector $\boldsymbol{m}_{a,\cdot,t}$ is closest to the $l$-th cluster center $\boldsymbol{\theta}_{l,t}$.

The problem with the above strategy is that it does not consider successive timestamps. To solve this issue, we propose a temporal landmark selection strategy. We re-formulate Eq.(5) as follows:

$$\underset{\{\Theta_{l,t}\}_{l=1}^{\varrho_t}}{\arg\min} \sum_{l=1}^{\varrho_t} \sum_{a \in \Theta_{l,t}} (\|\boldsymbol{m}_{a,\cdot,t} - \boldsymbol{\theta}_{l,t}\|_2^2 + \lambda \|\boldsymbol{m}_{a,\cdot,t-1} - \boldsymbol{\theta}_{l,t}\|_2^2), \tag{6}$$

where the second term ensures that the selected landmarks are still representative in consecutive timestamps, and $\lambda$ serves as a hyperparameter to balance the importance of these items. Fortunately, we can efficiently derive an analytical solution of Eq.(6) as follows:

$$\boldsymbol{\theta}_{l,t} = \begin{cases} \dfrac{\sum_{a \in \Theta_{l,t}} \boldsymbol{m}_{a,\cdot,t}}{|\Theta_{l,t}|}, & \text{if } t = 1, \\[3mm] \dfrac{\sum_{a \in \Theta_{l,t}} (1+\lambda)(\boldsymbol{m}_{a,\cdot,t}+\boldsymbol{m}_{a,\cdot,t-1})}{|\Theta_{l,t}|}, & \text{if } t > 1. \end{cases} \tag{7}$$

where $|\Theta_{l,t}|$ denotes the number of samples in the $l$-th cluster. By iteratively updating Eq.(7) until convergence, we assign the nearest $|U_t|/\varrho_t$ nodes to the $l$-th cluster center $\boldsymbol{\theta}_{l,t}$ ($l \in \{1, \ldots, \varrho_t\}$) to $U_t$, where $U_t$ denotes the temporal landmarks at the $t$-th timestamp and $|U_t|$ denotes the number of samples in $U_t$.

We then define the PMI matrix for temporal landmarks $|U_t|$ as $M_t^{00}$, which requires being factorized first as follows:

$$\mathcal{L}_t^{\text{Lm}} = \|M_t^{00} - \Phi_t \Psi_t\|_F^2, \quad s.t., \Phi_t \geq 0, \Psi_t \geq 0, \Phi_t \mathbf{1} = \mathbf{1}, \tag{8}$$

where $\Phi_t, \Psi_t$ are basis and coefficient matrices, respectively. These matrices are kept fixed during the following processes to serve as references, ensuring node embeddings' coherence and consistency.

**Randomly Subset Separation in Fig 2(b).** We randomly divide all nodes, except those selected as landmarks of $\mathcal{G}_t$, into $s$ subsets. We find that $s = 50$ is suitable for all large-scale dynamic graphs.

**Separated Matrix Factorization in Fig 2(c).** After dividing all nodes into $s$ subsets, Eq.(2) can be re-formulated as follow:

$$M_t = \begin{pmatrix} M_t^{11} & \cdots & M_t^{1s} \\ \vdots & \ddots & \vdots \\ M_t^{s1} & \cdots & M_t^{ss} \end{pmatrix} \approx \begin{pmatrix} C_t^1 H_t^1 & \cdots & C_t^1 H_t^s \\ \vdots & \ddots & \vdots \\ C_t^s H_t^1 & \cdots & C_t^s H_t^s \end{pmatrix}, \; s.t. \begin{bmatrix} C_t \geq 0 \\ H_t \geq 0, \\ C_t \mathbf{1} = \mathbf{1} \end{bmatrix} \tag{9}$$

where $M_t^{ii}$ repents intra-subset information within the $i$-th subset, while $M_t^{ij}$ captures inter-subset information between two subsets. Independently factorizing these matrices can result in a significant embedding drift between $C_t^i$ and $C_t^j$, i.e., nodes in these subsets may be projected into different hidden spaces with distinct basis vectors.

**Theorem 1.** *For $\forall i$ satisfying $1 \leq i \leq s$, assuming $C_t^i$ and $H_t^i$ in Eq.(9) can be linearly represented by the basis and coefficient matrices of the landmarks, i.e., $C_t^i = P_t^i \Phi_t$ and $H_t^i = \Psi_t Q_t^i$. Then, jointly considering the matrix factorization of the landmarks $M_t^{00}$ with each sub-matrix ensures embedding consistency between subsets of nodes.*

According to Theorem 1, to ensure embedding consistency of intra-subsets, intra-subsets matrix factorization is formulated as follows:

$$\mathcal{L}_t^{\text{intra}} = \sum_{i=1}^s (\|M_t^{ii} - C_t^i H_t^i\|_F^2 + \|M_t^{0i} - \Phi_t H_t^i\|_F^2 + \|M_t^{i0} - C_t^i \Psi_t\|_F^2) \tag{10}$$

$$= \sum_{i=1}^s (\|M_t^{ii} - P_t^i M_t^{00} Q_t^i\|_F^2 + \|M_t^{0i} - M_t^{00} Q_t^i\|_F^2 + \|M_t^{i0} - P_t^i M_t^{00}\|_F^2).$$

And inter-subsets matrix factorization can be formulated as follows:

$$\mathcal{L}_t^{\text{inter}} = \sum_{1 \leq i \leq s, i \neq j} \|M_t^{ij} - P_t^i M_t^{00} Q_t^i\|_F^2 + \|M_t^{ji} - P_t^j M_t^{00} Q_t^i\|_F^2. \tag{11}$$

Finally, the overall objective function can be formulated as follows:

$$O_t = \mathcal{L}_t^{\text{intra}} + \mathcal{L}_t^{\text{inter}} + \alpha \sum_{i=1}^s \|C_t^i - C_{t-1}^i\|_F^2, \; s.t. C_t^i, H_t^i \geq 0, C_t^i \mathbf{1} = \mathbf{1} \tag{12}$$

For the proof and optimization process, please refer to Appendix G.

## 4.2 Bi-clustering Regularization

Real-world dynamic graphs always contain much noise and irregular evolution patterns, directly obtaining communities from $C_t$ will be easily affected by noisy data (Figure 6). To improve robustness against noise and jointly optimize graph embedding and clustering, we introduce bi-clustering theory [47] as a regularization item into our overall objective function Eq.(12). To realize this goal, we first introduce the nuclear norm theory as follows.

**Theorem 2.** *Let $L_{S_t} = I - D^{-1/2} S_t D^{-1/2}$ be the normalized Laplacian matrix, where $D$ is the degree matrix of $S_t$. The multiplicity $k$ of the eigenvalue 0 of $L_{S_t} \in \mathbb{R}^{n \times n}$ is equal to the number of connected components of the bipartite graph $S_t = \begin{pmatrix} 0 & C_t \\ C_t^T & 0 \end{pmatrix}$, where $n$ denotes the dimension of $L_{S_t}$ and $T$ indicates the matrix transpose operation.*

Theorem 2 indicates that if $rank(L_{S_t}) = n - k$, $S_t$ has $k$ purity connected components (clusters), i.e., we need to minimize the $k$ smallest eigenvalues of $L_{S_t}$ to be 0. Suppose $\sigma_i(L_{S_t})$ is the $i$-th

smallest eigenvalue of $L_{S_t}$ and $\sigma_i(L_{S_t}) \geq 0$ since $L_{S_t}$ is positive semi-defined. Then, the issue can be formulated as $\sum_{i=1}^k \sigma_i(L_{S_t}) \approx 0$, however, optimizing this item is difficult. According to KyFan's Theorem [13], minimizing the sum of $k$ smallest eigenvalues can be transformed into an easy trace optimization issue, $\sum_{i=1}^k \sigma_i(L_{S_t}) \iff Tr(F_t^T L_{S_t} F_t)$, where $Tr()$ denotes the trace of the matrix and $F_t$ is a learnable parameter matrix with orthogonality constraint.

**Theorem 3.** *The bi-clustering regularization on the $i$-th subset $S_t^i$ is equal to the imposing constraints on $C_t^i$, i.e., when $S_t^i$ contains $k$ pure clusters, $C_t^i$ will also exhibit $k$ pure clusters. And bi-clustering regularization is decomposable, i.e., the constraint on the matrix $M_t$ is equal to the constraint on each of its node subsets.*

According to Theorem 3, we can add the bi-clustering regularization (Bcr) to each subset. Then, Eq.(12) can be re-formulated as follows:

$$O_t = \mathcal{L}_t^{\text{intra}} + \mathcal{L}_t^{\text{inter}} + \alpha \sum_{i=1}^s \|C_t^i - C_{t-1}^i\|_F^2 + \beta \mathcal{L}_t^{\text{Bcr}}, \tag{13}$$

$$s.t. C_t^i, H_t^i \geq 0, C_t^i \mathbf{1} = \mathbf{1}, \mathcal{L}_t^{\text{Bcr}} = \sum_{i=1}^s Tr((F_t^i)^T L_{S_t^i} F_t^i), (F_t^i)^T F_t^i = I,$$

where I is the identity matrix, and $\alpha, \beta$ are hyperparameters. Proof is shown in Appendix H.

## 4.3 Selective Embedding Updating

The item $\sum_{i=1}^s \|C_t^i - C_{t-1}^i\|_F^2$ in Eq. (13) ensures temporal smoothness between timestamps; however, the primary focus is on overall smoothness, and the fine-grained smoothness between individual node pairs is overlooked. This results in heterogeneity in node embedding between successive snapshots, thus severely undermining interpretability and visualizability during the analysis of dynamic community trajectories. To avoid this issue and further improve clustering efficiency and accuracy, we devise a fine-grained node-level temporal smoothing strategy. We first separating nodes into static and dynamic groups and then update only the embeddings of those dynamically changing nodes, while the embeddings of static nodes are fixed and shared between each timestamp.

Most nodes in dynamic graphs follow gradual and stable evolution patterns, maintaining their embeddings relatively unchanged over time [39]. The remaining dynamic nodes are defined as those whose topological structures undergo significant changes or whose positions shift considerably relative to dynamic landmarks.

Based on this assumption, dynamic nodes can be defined as:

$$\Delta \epsilon_{a,t} = \underbrace{\|w_{a,\cdot,t} - \overline{w}_{a,\cdot,t}\|_2^2}_{\text{topological changes}} + \underbrace{\|(w_{a,\cdot,t} - \theta_{a,t}) - (w_{a,\cdot,t} - \overline{\theta}_{a,t})\|_2^2}_{\text{relative positions shift}}, \tag{14}$$

where $w_{a,\cdot,t}$ is the $a$-th row of weighted adjacency matrix $W_t$, $\overline{w}_{a,\cdot,t}$ is the average among three successive snapshots: $\overline{w}_{a,\cdot,t} = (w_{a,\cdot,t-1} + w_{a,\cdot,t} + w_{a,\cdot,t+1})/3$, $\theta_{a,t}$ is the clustering center closest to $w_{a,\cdot,t}$, and $\overline{\theta}_{a,\cdot,t}$ is the averaged among three clustering centers closest to $w_{a,\cdot,t}$, i.e., $\overline{\theta}_{a,\cdot,t} = (\theta_{a,\cdot,t-1} + \theta_{a,\cdot,t} + \theta_{a,\cdot,t+1})/3$. When $t=1$ or $t=\tau$, we ignore $w_{a,\cdot,0}/\theta_{a,\cdot,0}$ and $w_{a,\cdot,\tau+1}/\theta_{a,\cdot,\tau+1}$, respectively. $\Delta \epsilon_{\cdot,t}$ can be considered as a threshold, measuring the dynamics of each node, to divide the nodes into a dynamic set $X_t$ with $\mu\%$ nodes and a static set $Y_t$.

We then fix the static node embeddings in $Y_t$ unchanged and only update the $\mu\%$ dynamic node embeddings in $X_t$. Then, landmarks

factorization of Eq.(8) can be re-formulated as follows:

$$\tilde{\mathcal{L}}_t^{\text{Lm}} = \left\| \begin{pmatrix} M_{xx,t}^{00} & M_{xy,t}^{00} \\ M_{yx,t}^{00} & M_{yy,t}^{00} \end{pmatrix} - \begin{pmatrix} \Phi_{x,t} \\ \Phi_{y,t} \end{pmatrix} (\Psi_{x,t}, \Psi_{y,t}) \right\|_{\text{F}}^2, \quad (15)$$

$$s.t. \quad \Phi_{y,t} = \Phi_{y,t-1}, \ \Psi_{y,t} = \Psi_{y,t-1},$$

where $M_{xx,t}^{00}$ and $M_{yy,t}^{00}$ represent the PMI matrix of static and dynamic nodes in $M_t^{00}$, respectively. $\Phi_{x,t}$ and $\Phi_{y,t}$ denote the sub-blocks of $\Phi_t$ for the dynamic and static landmarks, respectively.

Following Eq.(15), the overall objective function Eq.(13) can be factorized by only updating dynamic node embeddings while removing the temporal smoothing item, re-formulated as follows:

$$O_t = \tilde{\mathcal{L}}_t^{\text{intra}} + \tilde{\mathcal{L}}_t^{\text{inter}} + \beta \tilde{\mathcal{L}}_t^{\text{Bcr}}, \quad (16)$$

where $\tilde{\mathcal{L}}_t^{\text{intra}}$, $\tilde{\mathcal{L}}_t^{\text{inter}}$, and $\tilde{\mathcal{L}}_t^{\text{Bcr}}$ are the versions where the selective embedding updating has been applied.

By applying the constraint of fixing static node embeddings, we gain two advantages: **(i) it prevents updates to static node embeddings from introducing noise**, allowing us to better leverage historical information to improve model performance; **(ii) it allows us to remove the smoothness term** $\|C_t^i - C_{t-1}^i\|_{\text{F}}^2$ in Eq.(13), which significantly reduces computational cost. Please refer to Appendix I for a detailed definition and optimization of Eq.(16).

### 4.4 Complexity Analysis

**Time Complexity.** The time complexity of selecting $|U_t|$ landmarks with $\varrho_t$ clustering centers of all nodes $|V_t|$ using Eq.(7) is $O(|V_t|\varrho_t l_1)$, where $l_1$ is the number of iterations for converging to the optimal global solution. The time complexity of performing $\Phi_t$ and $\Psi_t$ by using Eq.(I.12) and Eq.(I.14) is $O(|U_t|^2 r l_2)$, where r is the number of dimensions and $l_2$ is the number of iterations to optimize Eq.(15) by gradient descent. The time complexity of updating $F_t^i$ in Eq.(I.27) for block $C_t^i$ with $|\Gamma_t^i|$ nodes is $O(|U_t|^3 + |U_t|^2|\Gamma_t^i|)$ [47]. Taking into account the above complexity, the total time complexity of using our method for dynamic graphs with $\tau$ timestamps is $O(|U_t|^3 + |U_t|^2|\Gamma_t^i| + |U_t|^2 r l_2 + |V_t|\varrho_t l_1) = O(\max\{|U_t|, |\Gamma_t^i|, rl_2\}|U_t|^2)$, while standard matrix factorization methods need time complexity of $O(|V_t|^3)$. Compared to standard matrix factorization methods, our method is more efficient.

**Space Complexity.** Since our method takes only snapshots $\mathcal{G}_{t-1}$, $\mathcal{G}_t$, and $\mathcal{G}_{t+1}$ as input to identify communities in the $t$-th timestamp, the space complexity is $O(|V_t|^2)$ including the space $O(|V_t|r)$ to store the matrices $C_t$ and $H_t$ with $r$ as the dimensions of matrix.

## 5 Experiment

**Datasets.** As shown in Table 1, following previous works [31, 72], we evaluate baselines on six synthetic dynamic graphs and five real-world dynamic graphs with varying number of nodes and edges. Synthetic dynamic graphs are generated following regular evolution rules. SYN-FIX/SYN-VAR [24] randomly exchange communities of some nodes. Green datasets [15] consider four evolution events including *Birth-Death*: existing communities are removed or generated by randomly selecting nodes from other communities; *Expand-Contract*: communities are expanded or contracted; *Hide*: communities are randomly hidden; *Merge-Split*: communities are split or merged. We also evaluate on various real-world domains,

**Table 1: Detailed statistics of dynamic graph benchmarks.**

| Dynamic Graphs | # of Nodes | # of Edges | # of Snapshots |
|---|---|---|---|
| SYN-FIX | 128 | 1,248,231 | 10 |
| SYN-VAR | 256 | 6,259,526 | 10 |
| Birth-Death | 30K & 100K | 12M & 24M | 10 & 20 |
| Expansion | 30K & 100K | 13M & 26M | 10 & 20 |
| Hide | 30K & 100K | 13M & 28M | 10 & 20 |
| Merge-Split | 30K & 100K | 14M & 29M | 10 & 20 |
| Wikipedia | 8,400 | 162,000 | 5 |
| Dublin | 11,000 | 415,900 | 5 |
| arXiv | 28,100 | 4,600,000 | 5 |
| Flickr | 2,302,925 | 33,100,000 | 5 |
| Youtube | 3,200,000 | 12,200,000 | 5 |

including *Academic Graphs*: arXiv [27]; *Social Graphs*: Dublin [19] and Flickr [43] and *Website Interaction Graphs*: Wikipedia [26] and Youtube [43]. Appendix C provides more details.

**Baselines.** We compare our method with 14 best-performing baselines, *i.e.,* Neural Network-based methods: CSEA [14], DSCPCD [66], SepNE [34], node2vec [17], LINE [62], RNNGCN [69], ROLAND [71], and TGC [38]; and Matrix Factorization-based methods: PisCES [35], DYNMOGA [15], NE2NMF [31], RTSC [72], RDMA [51], and jL-MDC [30]. Appendix D provides more details about these baselines.

**Implementation Details.** Following previous works [35, 42], we use normalized mutual information (NMI) [10] and normalized F1-score (NF1) [52] to measure clustering accuracy. We reproduced the baselines using their optimal parameters and reported average performance over five repeated runs with different random seeds. We conducted multiple t-tests with Benjamini-Hochberg [3] correction to assess the statistical significance of the performance. We took Birth-Death-30K as validation datasets for hyperparameter tuning. With grid search, our method achieves the best performance when number of blocks $s = 50$, dimension of embeddings $r = 1,000$, percentage of landmarks $\|U_t\| = 0.5$, percentage of dynamic nodes $\mu = 0.16$, $\lambda = 0.2$ in Eq.(6) and $\beta = 20$ in Eq.(16).

### 5.1 Performance Evaluation

The performance of various baselines in terms of NMI and NF1 scores on synthetic and real-world dynamic graphs is shown in Table 2. We observe that DyG-MF achieves the highest NMI and NF1 scores across all dynamic graphs. This can be attributed to its temporal separated matrix factorization, bi-clustering regularization, and selective embedding updating. Specifically, compared to the neural network-based methods RNNGCN and ROLAND, DyG-MF improves NMI scores by 5% and 3.8% on Flickr and Youtube, since DyG-MF jointly optimizes node embeddings and clustering, ensuring that node embeddings provide the most suitable features for clustering. In comparison with matrix factorization-based baselines, DyG-MF outperforms them by leveraging fine-grained temporal smoothness to capture dynamics at the node level (w/o SEU versus DyG-MF in Table 2) and utilizing bi-clustering regularization to reduce noise in real-world dynamic graphs (w/o BR versus DyG-MF in Table 2). Figure 3 further shows the performance of the baselines at each timestamp, showing that DyG-MF consistently outperforms baselines and can be effectively applied to real world.

Table 2: Overall performances on dynamic graphs. Bold and _Underline_ indicates the best and second-best performing methods. Symbol † indicates that DyG-MF significantly surpassed all baselines with a p-value< 0.005. The top eight methods are neural network-based methods, and the other methods are matrix factorization-based methods. ✗ means that it cannot be executed due to memory and running time constraints. DyG-MF w/o TSMF, w/o BR, and w/o SEU are introduced in Sec 5.4.

| Methods | SYN-FIX | | SYN-VAR | | Birth-30K | | Expand-30K | | Hide-100K | | Merge-100K | | Wikipedia | | Dublin | | arXiv | | Flickr | | Youtube | |
|---|---|---|---|---|---|---|---|---|---|---|---|---|---|---|---|---|---|---|---|---|---|---|
| | NMI | NF1 | NMI | NF1 | NMI | NF1 | NMI | NF1 | NMI | NF1 | NMI | NF1 | NMI | NF1 | NMI | NF1 | NMI | NF1 | NMI | NF1 | NMI | NF1 |
| CSEA [14] | 70.8 | 73.1 | 72.4 | 74.2 | 88.6 | 68.8 | 87.3 | 67.2 | 74.9 | 62.3 | 76.6 | 61.9 | 28.6 | 7.5 | 29.6 | 10.2 | 28.3 | 9.8 | 30.2 | 13.6 | 29.6 | 14.4 |
| DSCPCD [66] | 73.2 | 75.2 | 76.1 | 77.3 | 89.2 | 69.8 | 88.9 | 70.3 | 79.6 | 63.5 | 81.0 | 64.4 | 28.2 | 7.3 | 32.4 | 11.9 | 31.8 | 10.9 | 34.3 | 18.2 | 32.6 | 15.9 |
| SepNE [34] | 96.9 | 96.8 | 91.3 | 88.8 | 92.5 | _89.4_ | 92.1 | 81.8 | 89.0 | 81.1 | 89.2 | 78.6 | 31.4 | 9.8 | 49.2 | 21.0 | 42.8 | 25.8 | 40.3 | 22.4 | 39.5 | 21.0 |
| node2vec [17] | 98.3 | 97.9 | 92.6 | 91.3 | 93.8 | 85.8 | 93.2 | 83.6 | 91.2 | 85.6 | 89.0 | 78.2 | 33.5 | 10.2 | 50.3 | 23.2 | 44.3 | 26.8 | 42.4 | 25.5 | 42.8 | 24.5 |
| LINE [62] | 97.8 | 97.6 | 91.2 | 89.3 | 92.1 | 85.9 | 92.3 | 84.8 | 89.5 | 82.2 | 88.0 | 77.6 | 31.2 | 9.6 | 49.8 | 21.2 | 43.2 | 26.0 | 41.5 | 24.6 | 41.9 | 23.8 |
| RNNGCN [69] | 99.2 | 98.8 | 95.5 | 90.3 | 96.6 | 84.5 | 95.9 | 85.1 | 92.1 | 84.3 | 91.2 | 80.9 | 40.3 | _22.5_ | 52.2 | _31.6_ | 45.4 | 26.2 | _48.5_ | 30.2 | 47.6 | 32.3 |
| ROLAND [71] | 98.2 | 97.7 | 93.8 | 89.2 | 95.5 | 83.2 | 94.1 | 83.8 | _93.3_ | _85.8_ | _92.8_ | _81.6_ | 42.2 | 22.3 | _53.6_ | _31.6_ | _46.8_ | _27.6_ | 47.5 | 31.4 | _48.4_ | _33.2_ |
| TGC [38] | 98.3 | 97.9 | 93.5 | 90.6 | 95.3 | 83.0 | 93.8 | 83.6 | 92.8 | 85.4 | 91.5 | 81.2 | 41.3 | 22.3 | 52.8 | 31.3 | 45.8 | 26.8 | 47.8 | _31.6_ | 47.9 | 32.6 |
| PisCES [35] | 99.0 | 99.7 | 88.1 | 56.6 | 91.2 | 41.6 | 92.6 | 49.0 | ✗ | ✗ | ✗ | ✗ | 32.1 | 9.9 | 46.3 | 16.2 | 38.2 | 14.5 | ✗ | ✗ | ✗ | ✗ |
| DYNMOGA [15] | 92.5 | 95.6 | 84.2 | 61.6 | _98.1_ | 78.1 | _98.2_ | 65.3 | ✗ | ✗ | ✗ | ✗ | 36.2 | 9.9 | 49.8 | 20.1 | 39.1 | 24.5 | ✗ | ✗ | ✗ | ✗ |
| NE2NMF [31] | 97.8 | 95.9 | 94.2 | 93.6 | 97.1 | 76.1 | 97.5 | 63.1 | ✗ | ✗ | ✗ | ✗ | 34.1 | 8.2 | 47.9 | 18.9 | 38.2 | 22.9 | ✗ | ✗ | ✗ | ✗ |
| RTSC [72] | 99.2 | 99.0 | 98.7 | 92.8 | 92.8 | 55.3 | 92.1 | 53.2 | ✗ | ✗ | ✗ | ✗ | 30.6 | 11.3 | 46.6 | 19.3 | 38.2 | 20.2 | ✗ | ✗ | ✗ | ✗ |
| jLMDC [30] | _99.7_ | _99.9_ | 99.9 | _98.4_ | 98.0 | 77.4 | 97.6 | 66.6 | ✗ | ✗ | ✗ | ✗ | _44.6_ | 22.1 | 48.3 | 21.9 | 45.6 | 26.9 | ✗ | ✗ | ✗ | ✗ |
| RDMA [51] | 98.4 | 97.8 | 95.5 | 94.8 | 95.3 | 69.8 | 94.8 | _85.5_ | ✗ | ✗ | ✗ | ✗ | 33.8 | 10.2 | 47.2 | 18.6 | 41.6 | 25.2 | ✗ | ✗ | ✗ | ✗ |
| DyG-MF (Ours) | **100** | **100** | **100** | **100** | **99.9†** | **90.2†** | **99.2†** | **90.9†** | **94.3†** | **86.5†** | **94.4†** | **83.2†** | **50.4†** | **25.8†** | **56.1†** | **33.7†** | **51.8†** | **30.2†** | **52.3†** | **33.6†** | **51.8†** | **34.5†** |
| w/o TSMF | 100 | 100 | 100 | 100 | 99.9 | 90.3 | 99.3 | 90.9 | ✗ | ✗ | ✗ | ✗ | 50.6 | 25.9 | 56.4 | 33.9 | 52.0 | 30.5 | ✗ | ✗ | ✗ | ✗ |
| w/o BR | 99.0 | 99.5 | 88.9 | 71.8 | 97.3 | 78.2 | 97.8 | 82.6 | 90.8 | 84.5 | 88.7 | 78.1 | 45.8 | 21.5 | 52.1 | 30.6 | 45.8 | 25.8 | 48.2 | 29.6 | 48.5 | 31.6 |
| w/o SEU | 99.8 | 99.8 | 96.5 | 94.8 | 98.9 | 88.2 | 98.9 | 86.8 | 93.1 | 85.2 | 92.3 | 81.4 | 48.2 | 23.9 | 54.8 | 32.4 | 48.6 | 28.5 | 50.3 | 32.8 | 50.6 | 33.3 |

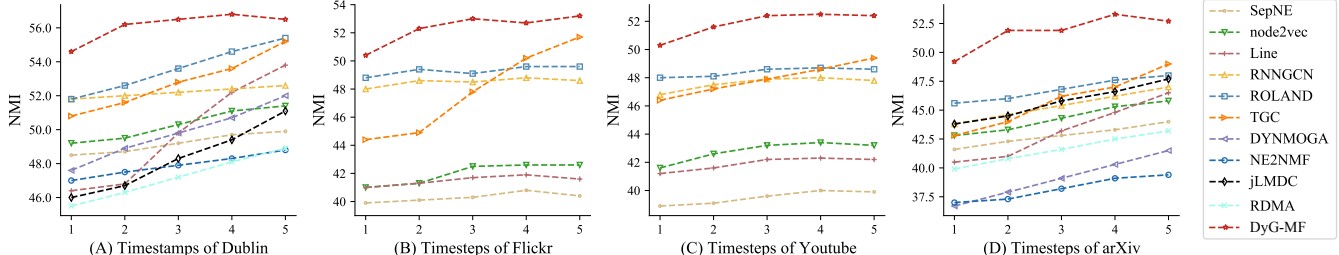

Figure 3: Performance on varying timestamps of selected best-performing baselines on four real-world datasets.

Moreover, we also employ two additional metrics, Modularity [45] and Density [7], to evaluate the quality of detected dynamic communities. This is necessary because NMI and NF1 rely on ground-truth labels, which can be easily affected by incorrect labeling. Specifically, Modularity evaluates the quality of inter-connections between nodes within a community, while Density measures outer-connections among communities without relying on labels. Figure 4 shows that DyG-MF outperforms three best-performing baselines on real-world dynamic graphs, indicating the effectiveness of DyG-MF in identifying high-quality communities.

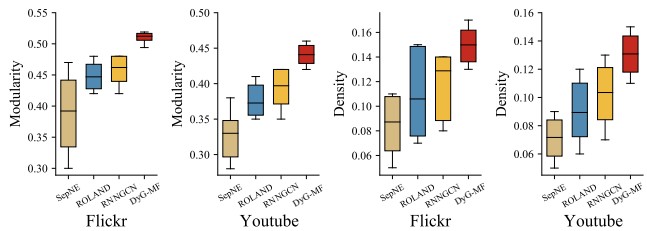

Figure 4: Modularity and Density on large dynamic graphs.

## 5.2 Scalability Evaluation

Table 3 shows the detailed running time of DyG-MF and baselines on large-scale dynamic graphs. Compared to the fastest baseline, SepNE, DyG-MF reduces the running time by 44.61% across all dynamic graphs, with a 52.27% reduction on synthetic dynamic graphs and 37.00% on real-world ones. To further investigate the scalability of DyG-MF, we conduct additional experiments on the Birth-Death dataset with varying numbers of snapshots and nodes, as shown in Figure 5. Specifically, DyG-MF's running time increases linearly as the number of snapshots and nodes grows, while the baselines show nearly exponential growth. This demonstrates DyG-MF's strong scalability for larger-scale real-world dynamic graphs, which can be attributed to its separated matrix factorization and selective embedding updating. Specifically, the temporal separated matrix factorization strategy breaks down the large-scale matrix factorization problem into smaller, more manageable subproblems without compromising clustering accuracy (jLMDC vs. DyG-MF in Tables 2 and 3). Moreover, compared to other matrix factorization-baselines like DYNMOGA, which update the embeddings of all nodes at each timestamp, DyG-MF updates only a small fraction of dynamically

**Table 3: Running Time for DyG-MF and baselines on large-scale synthetic and real-world dynamic graphs (sec). ✗ indicates that the corresponding methods could not be executed due to memory constraints or exceeded the time limit.**

| Methods | Synthetic dynamic graphs (↓) | | | | | Real-world dynamic graphs (↓) | | | | | |
|---|---|---|---|---|---|---|---|---|---|---|---|
| | Bir-Dea-100K | Expand-100K | Hide-100K | Mer-Spl-100K | *Avg.* | Wikipedia | Dublin | arXiv | Flickr | Youtube | *Avg.* |
| CSEA | 11,355 | 12,233 | 13,211 | 12,122 | 12,230 | 9,342 | 10,242 | 10,211 | 85,363 | 96,299 | 42,291 |
| DSCPCD | 10,255 | 11,323 | 10,232 | 11,211 | 10,755 | 8,882 | 9,323 | 9,299 | 82,242 | 94,233 | 40,795 |
| SepNE | 7,232 | 7,599 | 7,104 | 7,562 | 7,374 | 3,519 | 3,974 | 5,602 | 40,752 | 58,608 | 22,491 |
| LINE | 23,633 | 22,566 | 20,963 | 21,555 | 22,179 | 7,820 | 9,464 | 13,029 | 117,000 | 132,000 | 55,862 |
| node2vec | 16,963 | 17,070 | 17,799 | 17,705 | 17,384 | 5,474 | 7,098 | 10,032 | 99,450 | 121,440 | 48,698 |
| RNNGCN | 25,342 | 24,983 | 24,518 | 25,388 | 25,057 | 15,512 | 17,035 | 20,846 | 263,250 | 294,360 | 122,200 |
| ROLAND | 25,983 | 25,268 | 24,399 | 23,598 | 24,812 | 8,602 | 10,883 | 15,374 | 146,250 | 172,920 | 70,805 |
| TGC | 21,252 | 20,488 | 19,458 | 20,269 | 20,366 | 8,420 | 10,232 | 14,535 | 138,455 | 168,345 | 67,997 |
| PisCES | ✗ | ✗ | ✗ | ✗ | ✗ | 44,365 | 50,287 | 98,382 | ✗ | ✗ | ✗ |
| DYNMOGA | ✗ | ✗ | ✗ | ✗ | ✗ | 24,582 | 33,442 | 62,579 | ✗ | ✗ | ✗ |
| NE2NMF | ✗ | ✗ | ✗ | ✗ | ✗ | 39,482 | 44,377 | 79,255 | ✗ | ✗ | ✗ |
| RTSC | ✗ | ✗ | ✗ | ✗ | ✗ | 42,242 | 43,345 | 81,334 | ✗ | ✗ | ✗ |
| jLMDC | ✗ | ✗ | ✗ | ✗ | ✗ | 18,541 | 25,233 | 38,433 | ✗ | ✗ | ✗ |
| RDMA | ✗ | ✗ | ✗ | ✗ | ✗ | 33,482 | 48,257 | 66,598 | ✗ | ✗ | ✗ |
| DyG-MF | **3,434** | **3,523** | **3,425** | **3,693** | **3,518** | **1,829** | **2,304** | **2,717** | **32,460** | **40,752** | **16,012** |
| w/o TSMF | ✗ | ✗ | ✗ | ✗ | ✗ | 31,363 | 33,595 | 55,282 | ✗ | ✗ | ✗ |

evolving nodes (16%), showing its potential applicability in real-world scenarios. Figure 5(C-D) also confirms that higher efficiency and scalability do not reduce the NMI scores.

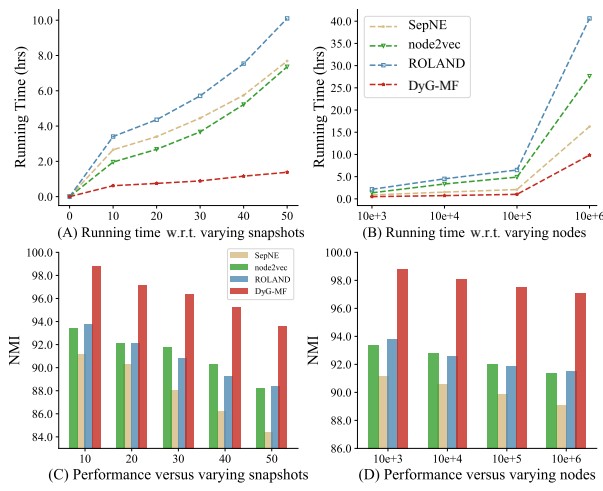

**Figure 5: Scalability w.r.t. varying snapshots and nodes.**

## 5.3 Robustness Evaluation

Real-world dynamic graphs often contain much noise and exhibit irregular evolution patterns. Table 2 and Figure 3 show that DyG-MF outperforms all baselines on real-world dynamic graphs, demonstrating its ability to filter out noise and capture more complex evolution patterns. To further support our statement, as shown in Figure 6, we contaminate dynamic graphs by adding 5%~30% noisy edges in each snapshot, following Tan et al. [61]. Compared to the best-performing baselines, DyG-MF shows a less performance degradation, indicating its robustness against temporal noisy edges. We also observe that w/o bi-clustering regularization significantly decreases the NMI score, showing that bi-clustering regularization serves as the main component of DyG-MF in maintaining robustness against noise attacks in dynamic graphs.

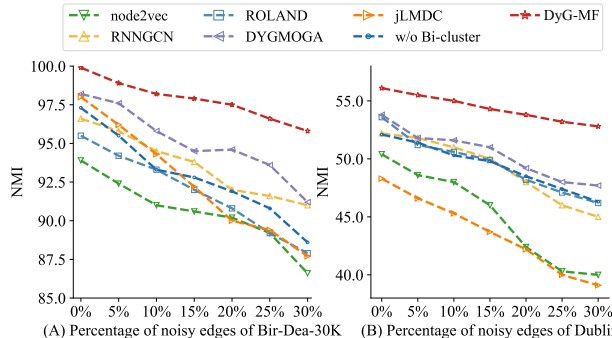

**Figure 6: Noise Attacks. NMI w.r.t. percentage of noisy edges.**

**Table 4: Ablation study on landmark selection strategies. Dynamic means landmarks are updated at each timestamp.**

| Strategies | Bir-Dea-30K | | Hide-30k | | Wikipedia | |
|---|---|---|---|---|---|---|
| DyG-MF + | NMI | NF1 | NMI | NF1 | NMI | NF1 |
| Fixed Random Selection | 90.2 | 83.2 | 89.2 | 80.1 | 42.2 | 19.5 |
| Fixed Greedy Selection | 92.6 | 85.5 | 92.1 | 82.3 | 44.8 | 21.3 |
| Fixed K-means Selection | 94.2 | 86.9 | 94.5 | 84.5 | 46.5 | 23.1 |
| Dynamic Random Selection | 88.2 | 81.5 | 90.3 | 82.3 | 41.8 | 17.9 |
| Dynamic Greedy Selection | 93.5 | 86.2 | 94.1 | 83.6 | 45.7 | 22.7 |
| Dynamic K-means Selection (Eq. (5)) | 96.3 | 87.5 | 96.2 | 86.1 | 47.5 | 23.8 |
| Our Selection (Eq. (6)) | **99.9** | **90.2** | **98.9** | **89.9** | **50.4** | **25.8** |

## 5.4 Ablation Study

We conduct an ablation study to evaluate the necessity of each component of DyG-MF. We consider the following three variants of DyG-MF: **(i) without Temporal Separated Matrix Factorization (w/o TSMF):** remove separated matrix factorization introduced in Sec 4.1 and replace it with Eq.(4); **(ii) without Bi-clustering Regularization (w/o BR):** remove bi-clustering regularization introduced in Sec 4.2; **(iii) without Selective Embedding Updating**

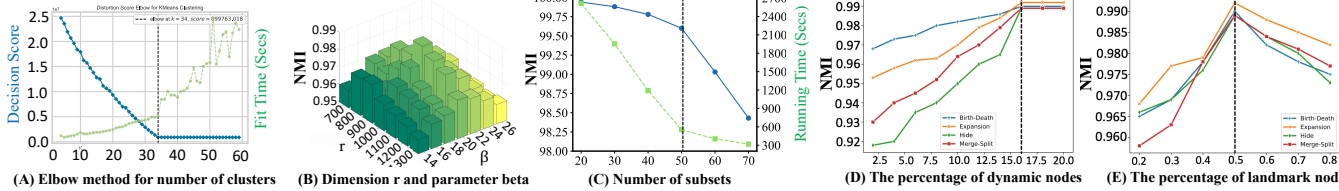

**Figure 7: (A)-(C) and show the number of clusters, and hyperparameter tuning of $r$, $\beta$ and $s$ on the first snapshot of Birth-Death-30K. (D)-(E) show the percentage of dynamic nodes and landmarks of four synthetic datasets with 30K nodes.**

**(w/o SEU):** remove fine-grained temporal smoothing strategy introduced in Sec 4.3. As shown in Table 2, removing any of these components negatively impacts overall performance on dynamic graph clustering, demonstrating their effectiveness and necessity.

To understand the role of temporal landmarks, as shown in Table 4, we selecte random sampling, greedy search [34] and $K$-means as potential strategies. We have three observations. **(i) Temporal landmarks selection is crucial**, as it can significantly affect performance on dynamic graph clustering. If landmarks cannot cover the entire feature space, there can be severe information loss for certain samples, leading to clustering errors. **(ii) Greedy sampling may not be as effective** as the $K$-means method, while it is faster. **(iii) Fixed setting underperforms the dynamic one** because the core landmarks will change over time. Thus, dynamically updating landmark selection can further improve performance.

## 5.5 Hyperparameters Analysis

We use the first snapshot of four synthetic event datasets as validation data to tune the hyperparameters $\{s, r, \beta, \mu, |U_t|\}$, where $s$ represents the number of separated subsets, $r$ is the dimension of the node embeddings, $\beta$ is the balanced parameter in Eq.(16), $\mu$ indicates the number of dynamic nodes, and $|U_t|$ refers to the number of landmarks. Following previous studies [69, 71], we adopt a grid search method to tune each hyperparameter while keeping the other parameters fixed. In Figure 7(A), the number of clusters $\varrho$ is automatically determined by the elbow method. Figure 7(B)-(C) shows that with $r = 1,000$, $\beta = 20$ and $s = 50$, DyG-MF achieves the highest NMI scores on the validation dataset. We do not display these parameters for the other three synthetic datasets, as they follow a similar trend. Figure 7(D)-(E) shows that when $\mu \in [16, 20]$ and $|U_t| \in [0.48, 0.52]$, DyG-MF achieves the best performance. Thus, we set $\mu = 16$ and $|U_t| = 0.5$ for the rest experiments. Note that for small-scale datasets like SYN-FIX/SYN-VAR, we set $s = 1$.

## 5.6 Case Study

To clearly demonstrate the effectiveness of DyG-MF, we present a visualization of the detected clusters using the t-SNE plot of the second snapshot from Wikipedia. As shown in Figure 8(A), the initialized node embeddings are randomly distributed in the two-dimensional space, without any discernible community structures. After optimizing by DyG-MF, we learn representative and suitable node embeddings that can automatically cluster nodes into distinct clusters, as illustrated in Figure 8(B).

To further illustrate the effectiveness, interpretability, and robustness of DyG-MF, we provide a case study using a Sankey plot

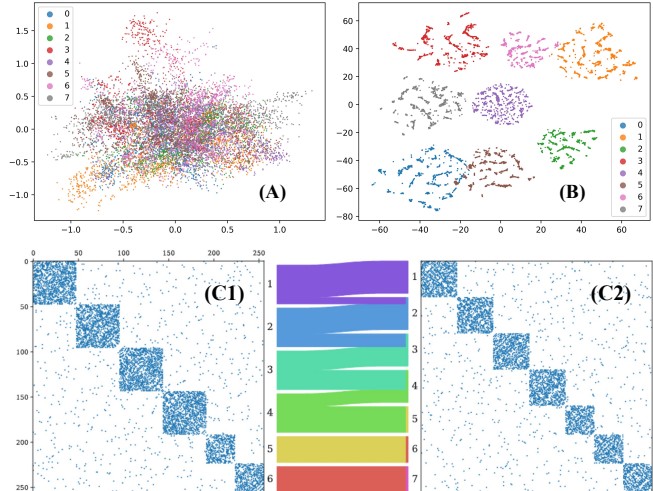

**Figure 8: t-SNE of the 2nd snapshot of Wikipedia: (A) initialized and (B) DyG-MF learned node embeddings. Communities of 3rd (C1) and 4th (C2) snapshots in SYN-VAR.**

to show community structures in the 3rd and 4th snapshots of the SYN-VAR, as shown in Figure 8(C1)-(C2). DyG-MF can effectively track the evolution patterns of individual node, *i.e.*, nodes from the second cluster in the 3rd snapshot are split into the second and third clusters in the 4th snapshot, highlighting DyG-MF's effectiveness and interpretability. Benefiting from the bi-clustering regularization, we can easily obtain the community evolution of nodes, where each diagonal block represents a community and the middle parts illustrate the transitions and changes between communities.

## 6 Conclusion

In this study, we proposed a novel scalable and robust temporal separated matrix factorization method to reveal the evolution mechanism of complex real-world complex systems. By jointly estimating graph embedding and clustering with Bi-clustering regularization and selective embedding updating, our method can achieve SOTA performance on synthetic and real-world dynamic graphs, illustrating its scalability, robustness, and effectiveness. In the future, we will design more separated matrix factorization strategies to preserve more global information, use incremental clustering to reduce time complexity during landmark selection, introduce diffusion models to enhance robustness, and extend our method to continuous-time dynamic graphs to enhance its flexibility.

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

# A  Limitations

The first limitation of this study is that DyG-MF only addresses non-overlapping clustering, while its performance on overlapping clustering remains underexplored. The second limitation is that DyG-MF has only been evaluated on large-scale real-world datasets containing up to 3,200,000 nodes, leaving its performance on even larger datasets still unexamined. Finally, with the advancement of natural language processing, many graph foundation models have been proposed. Exploring how to integrate these graph foundation models to obtain well-initialized node embeddings for improved performance is a promising area for future research.

# B  Pseudocode of DyG-MF

We give a Pseudocode of DyG-MF in Algorithm 1.

---

**Algorithm 1:** Pseudocode of our method.

**Input:** $\mathcal{G}_{\{1,\cdots,\tau\}}$: Dynamic Graphs;  $s, r, \beta, \mu, \lambda$: Hyperparameters.

**Output:** $\{V_l\}_{l=1}^{k_t}$ ($t \in \{1 \ldots, \tau\}$): Dynamic Communities.

1 **for** $t \in \{1, \ldots, \tau\}$ **do**

2      **Part I: Dynamic Graphs Separation and Processing.**

3      Randomly partition $\mathcal{V}_t$ into $s$ subsets;

4      Temporal landmarks selection of $U_t$ by Eq.(7);

5      Partition nodes into static set $Y_t$ and dynamic set $X_t$ by Eq.(14);

6      **Part II: Landmarks Matrix Factorization of Eq.(15)**

7      **repeat**

8          Update $\Phi_{x,t}$ by using Eq.(I.12);

9          Update $\Psi_{x,t}$ by using Eq.(I.14);

10      **until** *converge*;

11      **Part III: Separated Matrix Factorization of Eq.(16)**

12      **for** $i \in \{1, \ldots, s\}$ **do**

13          **repeat**

14              Fix other variables, update $F_{x,t}^i$ by using Eq.(I.27);

15              Fix other variables, update $Q_{x,t}^i$ by using Eq.(I.29);

16              Fix other variables, update $P_{x,t}^i$ by using Eq.(I.33);

17          **until** *converge*;

18          Calculate $C_t^i = [P_{x,t}^i \Phi_t; P_{y,t}^i \Phi_t]$

19      **end**

20      Recognizing clusters from $C_t^i$ for $\forall i$ satisfying $1 \leq i \leq s$;

21 **end**

---

# C  More Details about Datasets

We conducted experiments on 11 widely used datasets, including six synthetic and five real-world datasets, as shown in Table 1.

**SYN Datasets.** SYN-FIX and SYN-VAR were constructed with different dynamic settings for vertices and communities [24]. SYN-FIX fixes the number of communities at four and generates snapshots by randomly moving three vertices from each original community to new communities from the second to the final timestamp. In contrast, SYN-VAR consists of 256 vertices belonging to four equal-sized communities, randomly moving eight vertices from each of the four communities to form a new community with 32 vertices from the second to the fifth timestamp. The generated snapshots are then copied and reversed to create the final five snapshots.

**Green Datasets.** Considering the network sizes and the limited dynamic evolution of SYN-FIX/VAR, we generated four event-based temporal networks starting from the second timestamp [15]:

- **Birth-Death**: 5% existing communities are removed or generated by randomly selecting vertices from other communities;
- **Expansion**: 10% of communities are expanded or contracted by 50% of their original size;
- **Hide**: 10% of the communities are randomly hidden;
- **Merge-Split**: 20% of communities are split or merged in pairs.

We repeated the above process ($\tau$-1) times to construct the corresponding temporal networks, setting the number of timestamps to 10, the number of vertices to 30K/100K, the average degree of each snapshot at 100, the maximum degree at 200, the number of communities in the range [40, 60], and the mixing parameter to 0.2. As a result, we obtained Green datasets with 30K and 100K vertices, while we evaluated the evolutionary methods only on the 30K temporal networks.

**Real-world Datasets.** Following previous studies [31, 72], we conducted experiments on five widely used real-world temporal networks covering multiple applications. (1) **Academic graphs**: The arXiv dataset [? ] is a collaboration graph that describes the authors of scientific papers, covering papers from January 1993 to April 2003 (124 months) and consisting of 28,100 papers with 4,600,000 edges. (2) **Social networks**: The Dublin dataset [19] contains dynamic person-contact networks with 20-second intervals collected during the Infectious SocioPatterns event at the Science Gallery in Dublin. The Flickr dataset [43] is a dynamic social network with data collected over three months, featuring 950,143 new users and more than 9.7 million new links, focusing on how new links are formed. (3) **Website interaction networks**: The Wikipedia dataset [? ] is a bipartite editing network that contains temporal edits by users of Wikipedia pages. The Youtube dataset [43] includes a list of user-to-user links from the video-sharing website Youtube. To evaluate clustering accuracy, gold community labels are necessary for each vertex. We obtained gold community labels during the generation of synthetic temporal networks.

For real-world temporal networks, following previous studies [15] by aggregating all edges across all timestamps into a single graph and applying DYNMOGA to compute a soft modularity score Q [46], where the highest Q was considered the gold label for all vertices.

## D    More Details about Baselines

We compare our method with 14 best-performing baselines, which can be classified mainly into the following three classes:

**Coupling Baselines:**

- CSEA [14] first uses the Variational Autoencoder to reduce the dimension of the adjacency matrix and extracts the core structure of the coupling network. Then, $K$-means clustering is used to obtain information about the community structure.
- DSCPCD [66] detects community structures by maximizing the dual structural consistency of the coupling network, *i.e.*, the original explicit graph and the potential implicit graph have a consistent community structure.

**Two-stage Baselines:**

- SepNE [34] ignores the temporal information and estimates the clustering accuracy of separated matrix factorization in a proximity matrix from the given dynamic graphs. $K$-means is then used in the factorized matrix to obtain dynamic clusters.
- node2vec [17] uses a biased random walk procedure to explore neighborhoods in a breadth-first and depth-first sampling method so that neighborhood information can be maximally preserved. After obtaining graph embedding, $K$-means is used to capture dynamic clusters.
- LINE [62] is a breadth-first edge sampling method and considers both adjacent and deep interactions between vertices to learn graph embedding instead of using random walks. After obtaining graph embedding, $K$-means is used to capture dynamic clusters.
- RNNGCN [69] uses an RNN to learn the decay rates of each edge over timestamps to characterize the importance of historical information for current clustering. A two-layer graph convolutional network is used for dynamic graph clustering.
- ROLAND [71] extends the GNN to dynamic scenes by viewing the node representation at different layers as hierarchical node states and using GRUs to update these hierarchical vertex states based on newly observed vertices and edges.
- TGC [38] propose a general framework for deep Temporal Graph Clustering, which introduces deep clustering techniques to suit the interaction sequence-based batch-processing pattern of temporal graphs. They then discuss differences between temporal graph clustering and static graph clustering from several levels.

**Evolutionary Baselines:**

- PisCES [35] globally estimates and optimizes clustering drift on all snapshots. It uses non-negative matrix factorization, which is equal to spectral clustering, for dynamic graph clustering.
- DYNMOGA [15] estimates the clustering drift by minimizing the NMI between the community structures detected between two successive snapshots. And it maximizes clustering precision by directly decomposing the adjacency matrix and uses $K$-means for dynamic graph clustering.
- NE2NMF [31] uses previous and current snapshots to characterize cluster drift and locally optimizes drift at each timestamp. After decomposing the adjacency matrix by NMF to obtain a vertex representation matrix, it continues to detect communities by decomposing the vertex representation matrix.
- RTSC [72] uses the previous, current, and subsequent graphs to measure clustering drift. It applies non-negative matrix factorization to the common feature matrix of three successive graphs to estimate clustering precision. Finally, RTSC uses $K$-means as post-processing for dynamic graph clustering.
- jLMDC [30] propose a novel joint learning model for dynamic community detection through joint feature extraction and clustering. This model is formulated as a constrained optimization problem. Vertices are classified into dynamic and static groups by exploring the topological structure of temporal networks to fully exploit their dynamics at each time step. Then, jLMDC updates the features of dynamic vertices by preserving features of static ones during optimization. The advantage of jLMDC is

that the features are extracted under the guidance of clustering, promoting performance, and saving running time.

- RDMA [51] propose the robust memetic method and use the idea to optimize the detection of dynamic communities in complex networks. They work with dynamic data that affect the two main parts of the initial population value and the calculation of the evaluation function of each population, and there is no need to determine the number of communities in advance.

## E   Introduction of Criteria and Significant Tests

Following previous studies [35, 42], we used normalized mutual information (NMI) [10] and normalized F1 score (NF1) [52] to measure the precision of clustering. Let $k^*$ and $k$ be the gold and the predicted number of communities. We denote a confusion matrix $N \in \mathbb{R}^{k^* \times k}$, where the rows correspond to the gold communities $\{\mathcal{V}_l^*\}_{l=1}^{k^*}$, columns correspond to the predicted communities $\{\mathcal{V}_l\}_{l=1}^{k}$, and $n_{ij}$ is the number of vertices overlapped between the $i$-th real and $j$-th predicted communities. NMI can be formulated

$$\text{NMI} = \frac{-2 \sum_{i=1}^{k^*} \sum_{j=1}^{k} n_{ij} \log(\frac{n_{ij}|N|}{|\mathbf{n}_{i.}||\mathbf{n}_{.j}|})}{\sum_{i=1}^{k^*} |\mathbf{n}_{i.}| \log(\frac{|\mathbf{n}_{i.}|}{|N|}) + \sum_{j=1}^{k} |\mathbf{n}_{.j}| \log(\frac{|\mathbf{n}_{.j}|}{|N|})},$$

where the sum of all elements of $N$ is denoted as $|N|$, sum over the $i$-row $\mathbf{n}_{i.}$ and $j$-th column $\mathbf{n}_{.j}$ are denoted as $|\mathbf{n}_{i.}|$ and $|\mathbf{n}_{.j}|$.

Before introducing NF1, we first define that Precision is the percentage of vertices in $\mathcal{V}_i$ labelled as $\mathcal{V}_j^*$:

$$\text{Precision}(\mathcal{V}_i, \mathcal{V}_j^*) = \frac{|\mathcal{V}_i \cap \mathcal{V}_j^*|}{|\mathcal{V}_i|} \in [0, 1], \tag{17}$$

and Recall is the percentage of vertices in $\mathcal{V}_j^*$ covered by $\mathcal{V}_i$:

$$\text{Recall}(\mathcal{V}_i, \mathcal{V}_j^*) = \frac{|\mathcal{V}_i \cap \mathcal{V}_j^*|}{|\mathcal{V}_j^*|} \in [0, 1]. \tag{18}$$

Following Rossetti et al. [54], we combine precision and recall into their harmonic mean to define the F1 score as

$$\text{F1}(\mathcal{V}_i, \mathcal{V}_j^*) = 2 \frac{\text{Recall}(\mathcal{V}_i, \mathcal{V}_j^*) * \text{Precision}(\mathcal{V}_i, \mathcal{V}_j^*)}{\text{Recall}(\mathcal{V}_i, \mathcal{V}_j^*) + \text{Precision}(\mathcal{V}_i, \mathcal{V}_j^*)} \in [0, 1],$$

where we then averaged $\text{F1}(\mathcal{V}_i, \mathcal{V}_j^*)$ across all the identified pairs between the $\{\mathcal{V}_l^*\}_{l=1}^{k^*}$ and $\{\mathcal{V}_l\}_{l=1}^{k}$.

Following Rossetti [52], NF1 can be defined as

$$\text{NF1} = \frac{\text{F1} * \text{Coverage}}{\text{Redundancy}} \in (0, 1], \tag{19}$$

where Coverage identified the percentage of communities in $\mathcal{V}^*$ that were matched by at least one object in $\mathcal{V}$:

$$\text{Coverage} = \frac{|\mathcal{V}_{id}^*|}{|\mathcal{V}^*|} \in [0, 1], \tag{20}$$

where $\mathcal{V}_{id}^*$ was the subset of communities in $\mathcal{V}^*$ matched by $\mathcal{V}$. Redundancy is the percentage of unmatched real communities:

$$\text{Redundancy} = \frac{|\mathcal{V}|}{|\mathcal{V}_{id}^*|} \in [1, +\infty). \tag{21}$$

**Benjamini-Hochberg Correction** (*B-H*) is a powerful tool that decreases the false discovery rate [3]. Considering the reproducibility of the multiple significant test, we introduce how we adopt the *B-H* correction and give the hyperparameter values that we used.

We first performed a t-test with the default parameters[2] to calculate the p-value between each of the comparison methods with our method. We then put the individual p-values in ascending order as input to calculate the corrected p-value using the *B-H* correction. We directly use the "***multipletests(\*args)***" function from Python package[3] and set the hyperparameter of the false discovery rate $Q = 0.05$, which is a widely used default value. Finally, we obtain a cutoff value as the output of the ***multipletests*** function, where cut-off is a dividing line that distinguishes whether two groups of data are significant. If the p-value is smaller than the cut-off value, we can conclude that two groups of data are significantly different.

## F   Extension of DyG-MF in Complex Scenarios

**Table 5: Performance with invisible subsequent graphs and varying number of total nodes setting.**

| Methods | Dublin | | arXiv | | Flickr | | Youtube | |
|---|---|---|---|---|---|---|---|---|
| | NMI | NF1 | NMI | NF1 | NMI | NF1 | NMI | NF1 |
| RNNGCN | 52.2 | 31.6 | 45.4 | 26.2 | 48.5 | 30.2 | 47.6 | 32.3 |
| ROLAND | 53.6 | 31.6 | 46.8 | 27.6 | 47.5 | 31.4 | 48.4 | 33.2 |
| TGC | 52.8 | 31.3 | 45.8 | 26.8 | 47.8 | 31.6 | 47.9 | 32.6 |
| DyG-MF (w/o sub) | 54.2 | 31.8 | 50.2 | 28.7 | 49.5 | 31.6 | 49.9 | 33.6 |
| DyG-MF (Original) | **56.1** | **33.7** | **51.8** | **30.2** | **52.3** | **33.6** | **51.8** | **34.5** |

For any given dynamic graph, DyG-MF initializes the set of vertices $V_1$ based on the first snapshot. In the following timestamps, DyG-MF determines the current timestamp as the final one and computes Eq.(14) using only historical information $\mathbf{w}_{a.,t-1}$ and current information $\mathbf{w}_{a.,t}$. Meanwhile, for newly added vertices, we randomly initialize their representations and group them into a new block and merge this block with $M_{t-1}$ to construct the current $M_t$. To account for vertices that exist in the previous snapshot but disappear in the current snapshot, we merge blocks that fill with zeros for the vanishing vertices, ensuring that $M_{t-1}$ and $M_t$ maintain the same size. We also construct $W_t$ in a similar way. We consider newly added vertices as both landmarks and dynamic vertices, updating their representations with our overall objective equation in Eq.(16). We denote this configuration as "Ours (w/o sub)". As shown in Table 5, our method outperforms Ours (w/o sub) because the additional block for newly appearing vertices increases the number of blocks $s$, which results in less preserved landmark information. In addition, using only previous and current snapshots provides less information for selecting dynamic vertices compared to using three successive snapshots. Although Ours (w/o sub) performs worse than our method in Table 5, Ours (w/o sub) achieves the highest NMI and NF1 compared to other baselines. This result indicates the effectiveness of Ours (w/o sub) and its potential for separated matrix factorization.

---

[2]https://docs.scipy.org/scipy.stats.ttest_ind.html
[3]https://www.statsmodels.org/statsmodels.stats.multitest.multipletests.html

# G  Optimization for Eq.(12)

We first list the $i$-th sub-issue of optimizing Eq.(12) as:

$$O_{i,t} = \mathcal{L}_{i,t}^{\text{intra}} + \mathcal{L}_{i,t}^{\text{inter}} + \alpha \|C_t^i - C_{t-1}^i\|_{\text{F}}^2. \tag{G.1}$$

We follow the step of Li et al. [34] to estimate $\Phi_t$ and $\Psi_t$ in advance. Considering that the time complexity for SVD is $O(n^3)$ which is time-cost, we conduct $\Phi_t$ and $\Psi_t$ with gradient descent instead. Based on the Karush-Kuhn-Tucker (KKT) condition [23], $\Phi_t$ and $\Psi_t$ in Eq.(8) are updated iteratively at each timestamp as

$$\Phi_t \leftarrow \Phi_t \cdot \frac{M_t^{00}(\Psi_t)^T}{\Phi_t \Psi_t (\Psi_t)^T}, \tag{G.2}$$

$$\Psi_t \leftarrow \Psi_t \cdot \frac{(\Phi_t)^T M_t^{00}}{(\Phi_t)^T \Phi_t \Psi_t}. \tag{G.3}$$

By giving the converged steps $\Phi_t$ and $\Psi_t$, we iteratively update $Q_t^i$ and $P_t^i$ steps. We fix $P_t^i$ to deduce Eq.(G.1) as:

$$O_{i,t} = \|M_t^{ii} - P_t^i M_t^{00} Q_t^i\| + \|M_t^{0i} - M_t^{00} Q_t^i\| + \sum_{j \neq i} \|M_t^{ji} - M_t^{j0} Q_t^i\|. \tag{G.4}$$

The update rule for $Q_t^i$ is then conducted as:

$$Q_t^i \leftarrow Q_t^i \cdot \frac{(M_t^{00})^T J_1 + \sum_{j \neq i} (M_t^{j0})^T M_t^{ji}}{(M_t^{00})^T J_2 M_t^{00} Q_t^i + \sum_{j \neq i} (M_t^{j0})^T M_t^{j0} Q_t^i}, \tag{G.5}$$

where $J_1 = (P_t^i)^T M_t^{ii} + M_t^{0i}$, and $J_2 = I + (P_t^i)^T P_t^i$.

We also fix $Q_t^i$ to deduce Eq.(G.1) as:

$$O_{i,t} = \|M_t^{ii} - P_t^i M_t^{00} Q_t^i\| + \sum_{j \neq i} \|M_t^{ij} - P_t^i M_t^{0j}\| + \|M_t^{i0} - P_t^i M_t^{00}\| + \alpha \|P_t^i \Phi_t - P_{t-1}^i \Phi_t\|. \tag{G.6}$$

The update rule for $P_t^i$ is then conducted as

$$P_t^i \leftarrow P_t^i \cdot \frac{J_3 (M_t^{00})^T + \sum_{j \neq i} M_t^{ij} (M_t^{0j})^T + \alpha (P_{t-1}^i \Phi_t \Phi_t^T)}{P_t^i M_t^{00} J_4 + \sum_{j \neq i} P_t^i M_t^{0j} (M_t^{0j})^T + \alpha (P_t^i \Phi_t \Phi_t^T)}, \tag{G.7}$$

where $J_3 = M_t^{ii}(Q_t^i)^T + M_t^{i0}$, and $J_4 = Q_t^i (M_t^{00} Q_t^i)^T + (M_t^{00})^T$.

# H  Proof for Theorems

## H.1  Proof for Theorem 1

**Theorem 1** *For $\forall i$ satisfying $1 \leq i \leq s$, assuming $C_t^i$ and $H_t^i$ in Eq.(9) can be linearly represented by the basis and coefficient matrices of the landmarks, i.e., $C_t^i = P_t^i \Phi_t$ and $H_t^i = \Psi_t Q_t^i$. Then, jointly considering the matrix factorization of the landmarks $M_t^{00}$ with each sub-matrix ensures embedding consistency between subsets of nodes.*

PROOF.  We begin by considering the matrix factorization of the node subsets $M_t^{ij}$ and $M_t^{pq}$ as follows:

$$\|M_t^{ij} - C_t^i H_t^j\|_{\text{F}}^2. \quad \|M_t^{pq} - C_t^p H_t^q\|_{\text{F}}^2. \tag{H.1}$$

Here, $C_t^i$ and $C_t^p$ represent the node embeddings of the subsets $i$ and $p$, respectively. It is evident that the embeddings of $C_t^i$ and $C_t^p$ are independent, and there is no constraint enforcing them to reside in the same latent space. Without such a constraint, the embeddings learned for the different subsets are disjoint and inconsistent, resulting in discontinuities between them and undermining the overall quality of the embeddings. To resolve this issue, we impose conditions $C_t^i = P_t^i \Phi_t$ and $H_t^i = \Psi_t Q_t^i$, where $\Phi_t$ and $\Psi_t$ act as shared basis matrices for embedding consistency across all subsets of nodes. Under these constraints, the matrix factorization for $M_t^{ij}$ and $M_t^{pq}$ becomes:

$$\|M_t^{ij} - P_t^i \Phi_t ((Q_t^j)^T \Psi_t^T)^T\|_{\text{F}}^2. \quad \|M_t^{pq} - P_t^p \Phi_t ((Q_t^q)^T \Psi_t^T)^T\|_{\text{F}}^2. \tag{H.2}$$

$\Phi_t$ serves as a shared basis for the hidden space, and $P_t^i$ and $P_t^p$ are coefficient matrices that describe the node embeddings in terms of linear combinations of the basis vectors in $\Phi_t$. Consequently, the embeddings of different node subsets are aligned to the same latent space, ensuring that they are consistent across subsets. Additionally, $\Psi_t$ acts as a transformation matrix that projects the node embeddings of each subset into a common hidden space, thus further enforcing consistency between the embeddings of different subsets. Thus, by jointly considering the matrix factorization of the landmarks $M_t^{00}$ along with each sub-matrix $M_t^{ij}$ and $M_t^{pq}$, we guarantee that the node embeddings across different subsets are consistent and reside in the same latent space. This proves that embedding consistency is achieved between subsets of nodes.

$\square$

## H.2 Proof for Theorem 2

**Theorem 2** *Let* $L_{S_t} = I - D^{-1/2}S_t D^{-1/2}$ *be the normalized Laplacian matrix, where* $D$ *is the degree matrix of* $S_t$. *The multiplicity* $k$ *of the eigenvalue 0 of* $L_{S_t} \in \mathbb{R}^{n \times n}$ *is equal to the number of connected components of the bipartite graph* $S_t = \begin{pmatrix} 0 & C_t \\ C_t^T & 0 \end{pmatrix}$, *where* $n$ *denotes the dimension of* $L_{S_t}$ *and* $T$ *indicates the matrix transpose operation.*

Proof. To proof **Theorem 2**, we need the following prior lemma.

**Lemma 1.** *Let* $C_t \in \mathbb{R}^{n \times r}$, $S_t \in \mathbb{R}^{n \times n}$ *and* $L_{S_t} \in \mathbb{R}^{(n+r) \times (n+r)}$. *For any vector matrix* $F \in \mathbb{R}^{(n+r) \times (n+r)}$ *with* $\mathbf{f} \in \mathbb{R}^n$ *as its row vector, we have* $F^{\mathrm{T}} L_{S_t} F = \frac{1}{2} \sum_{a=1}^{n+r} \sum_{b=1}^{n+r} \|\mathbf{f}_{a.} - \mathbf{f}_{b.}\|_2^2 s_{ab}$.

For any matrix $F$, we can derive the following formula:

$$Tr(F^T L_{S_c} F) = Tr(F^T D_{S_c} F) - Tr(F^T S_c F)$$

$$= Tr([\mathbf{f}_{1.}^T, \cdots, \mathbf{f}_{\delta.}^T] \begin{bmatrix} d_{11} & 0 & \cdots & 0 \\ 0 & d_{22} & \cdots & 0 \\ \vdots & \vdots & \ddots & \vdots \\ 0 & 0 & \cdots & d_{\delta\delta} \end{bmatrix} \begin{bmatrix} \mathbf{f}_{1.} \\ \vdots \\ \mathbf{f}_{\delta.} \end{bmatrix}) - Tr([\mathbf{f}_{1.}^T, \cdots, \mathbf{f}_{\delta.}^T] \begin{bmatrix} s_{11} & s_{12} & \cdots & s_{1\delta} \\ s_{21} & s_{22} & \cdots & s_{2\delta} \\ \vdots & \vdots & \ddots & \vdots \\ s_{\delta 1} & s_{\delta 2} & \cdots & s_{\delta\delta} \end{bmatrix} \begin{bmatrix} \mathbf{f}_{1.} \\ \vdots \\ \mathbf{f}_{\delta.} \end{bmatrix})$$

$$= Tr([\mathbf{f}_{1.}^T d_{11}, \cdots, \mathbf{f}_{\delta.}^T d_{\delta\delta}]) \begin{bmatrix} \mathbf{f}_{1.} \\ \vdots \\ \mathbf{f}_{\delta.} \end{bmatrix}) - Tr([\mathbf{f}_{1.}^T s_{11} + \cdots + \mathbf{f}_{\delta.}^T s_{\delta 1}, \cdots, \mathbf{f}_{1.}^T s_{1\delta} + \cdots + \mathbf{f}_{1.}^T s_{\delta\delta}]) \begin{bmatrix} \mathbf{f}_{1.} \\ \vdots \\ \mathbf{f}_{\delta.} \end{bmatrix})$$

$$= \sum_{a=1}^{\delta} \mathbf{f}_{a.} d_{aa} \mathbf{f}_{a.}^T - \sum_{a=1}^{\delta} \sum_{b=1}^{\delta} \mathbf{f}_{b.} s_{ab} \mathbf{f}_{a.}^T$$

$$= \frac{1}{2} (2 \sum_{a=1}^{\delta} \mathbf{f}_{a.} d_{aa} \mathbf{f}_{a.}^T - 2 \sum_{a=1}^{\delta} \sum_{b=1}^{\delta} \mathbf{f}_{b.} s_{ab} \mathbf{f}_{a.}^T)$$

$$= \frac{1}{2} (\sum_{a=1}^{\delta} \sum_{b=1}^{\delta} s_{ab} \mathbf{f}_{a.}^2 - 2 \sum_{a=1}^{\delta} \sum_{b=1}^{\delta} \mathbf{f}_{a.} s_{ab} \mathbf{f}_{b.}^T + \sum_{b=1}^{\delta} \sum_{a=1}^{\delta} s_{ab} \mathbf{f}_{b.}^2)$$

$$= \frac{1}{2} \sum_{a=1}^{\delta} \sum_{b=1}^{\delta} \|\mathbf{f}_{a.} - \mathbf{f}_{b.}\|_2^2 s_{ab} \tag{H.3}$$

Since the Laplacian matrix $L_{S_t}$ is symmetric and positive semi-definite [47], its determinant is 0. If $\mathbf{f}$ is the all-ones vector $\mathbf{1} \in R^{n+r}$, we can then obtain:

$$L_{S_t} \mathbf{f} = L_{S_t} \mathbf{1} = (D - S_t)\mathbf{1} = [\mathbf{d}_1 \dots \mathbf{d}_n]^T - [\sum_{j=1}^{n} s_{1j} \cdots \sum_{j=1}^{n} s_{nj}]^T = 0 \tag{H.4}$$

Thus, $\mathbf{1}$ is the eigenvector corresponding to the eigenvalue 0. Since $L_{S_t}$ is positive semi-definite, its eigenvalue is non-negative, and the minimum eigenvalue of $L_{S_t}$ is 0, with $\mathbf{1}$ as the corresponding eigenvector.

The theorem will be proved for two cases below:

(i) **Case 1**: k=1, *i.e.,* $\mathcal{G}$ **is a connected graph.**

We need to prove that $L_{S_t}$ has only one eigenvalue of 0. Let $\mathbf{f}$ be the eigenvector corresponding to eigenvalue 0. Then, we have the following equation:

$$0 = \mathbf{f}^{\mathrm{T}} L_{S_t} \mathbf{f} = \frac{1}{2} \sum_{i=1}^{n} \sum_{j=1}^{n} w_{ij} (f_i - f_j)^2 \tag{H.5}$$

This implies $f_i = f_j$. This equality is transmitted alone all connected paths, meaning all elements of the eigenvector $\mathbf{f}$ must be equal and are in the span of the vector whose elements are all 1. Consequently, no other eigenvector corresponding to the eigenvalue 0. The prood is complete.

(ii) **Case 2: k > 1, we need to prove that** $L_{S_t}$ **has 0 eigenvalue with multiplicity k.**

The nodes are numbered according to the connected subgraphs. Since there are no edges between different connected subgraphs, the Laplace matrix $\mathbf{L}_{S_t}$ has a block structure:

$$
\begin{bmatrix}
L_1 & 0 & \cdots & 0 \\
0 & L_2 & \cdots & 0 \\
\vdots & \vdots & \ddots & \vdots \\
0 & 0 & \cdots & L_k
\end{bmatrix}
\tag{H.6}
$$

Each submatrix $L_i$ is also a Laplacian matrix for its connected component. By the same argument as in Case 1, the spectrum of $L_i$ is the union of the spectra of the $L_i$'s. The eigenvector corresponding to the eigenvalue 0 of $L_{S_i}$ is the eigenvector of $L_i$, with all other positions filled with 0. Specifically, the eigenvector corresponding to the eigenvalue of the vertices of the $i$-th connected components is 1 at the $i$-th component and 0 elsewhere, represented as $[0 \ldots 0\, 1 \ldots 1\, 0 \ldots 0]$.

Since each $L_i$ is a Laplacian matrix for a connected component, its 0 eigenvalue has multiplicity 1. Therefore, the number of linearly independent eigenvectors corresponding to the 0 eigenvalue of $L_{S_t}$ is equal to the number of connected components, and these eigenvectors are the indicator vectors of the connected components.

□

## H.3 Proof for Theorem 3

**Theorem 3** *The bi-clustering regularization on the $i$-th subset $S_t^i$ is equal to the imposing constraints on $C_t^i$, i.e., when $S_t^i$ contains $k$ pure clusters, $C_t^i$ will also exhibit $k$ pure clusters. And bi-clustering regularization is decomposable, i.e., the constraint on the matrix $M_t$ is equal to the constraint on each of its subsets.*

PROOF. We can separate the bi-clustering item $\mathcal{L}_{i,t}^{\text{Bcr}}$ in Eq.(16) based on dynamic and static nodes. Specifically, we further deduce the following relation between clustering structure $S_{x,t}^i$ and landmark-related matrix $S_{P_x,t}^i$, as

$$
S_{x,t}^i = \begin{pmatrix} 0 & C_{x,t}^i \\ (C_{x,t}^i)^T & 0 \end{pmatrix} = \begin{pmatrix} I & 0 \\ 0 & \Phi_t^T \end{pmatrix} \underbrace{\begin{pmatrix} 0 & P_{x,t}^i \\ (P_{x,t}^i)^T & 0 \end{pmatrix}}_{S_{P_x,t}^i} \begin{pmatrix} I & 0 \\ 0 & \Phi_t \end{pmatrix}.
\tag{H.7}
$$

Since the left and right matrices of $S_{P_x,t}^i$ are scalars, the trace optimization of $C_{x,t}^i$ is proportional to the matrix $P_{x,t}^i$, i.e.,

$$
Tr((F_t^i)^T L_{S_t^i} F_t^i) \propto Tr((F_{x,t}^i)^T L_{S_{x,t}^i} F_{x,t}^i) \propto Tr((F_{x,t}^i)^T L_{S_{P_x,t}^i} F_{x,t}^i),
\tag{H.8}
$$

□

## I Optimization for Overall Objective Function Eq. (16)

To further reduce running time and improve performance, we focus on the time-varying features of dynamic nodes and only update their node embeddings. The embeddings for the static nodes are fixed and are ignored for optimization. Therefore, optimization can be further facilitated due to fewer active nodes. We permute and divide the diagonal and off-diagonal blocks in Eq.(9), $M_t^{ii}$ and $M_t^{ij}$, into four sub-blocks:

$$
M_t^{ii} = \begin{pmatrix} M_{xx,t}^{ii} & M_{xy,t}^{ii} \\ M_{yx,t}^{ii} & M_{yy,t}^{ii} \end{pmatrix}, \quad M_t^{ij} = \begin{pmatrix} M_{xx,t}^{ij} & M_{xy,t}^{ij} \\ M_{yx,t}^{ij} & M_{yy,t}^{ij} \end{pmatrix},
\tag{I.1}
$$

where $M_{xy,t}^{ij} \in \mathbb{R}^{|X_t^i| \times |Y_t^j|}$ denotes the sub-block of $M_t^{ij}$ associated with dynamic nodes in $\Gamma_t^i$ and static ones in $\Gamma_t^j$. Similarly, the landmark information in Eq.(8) is re-formulated as:

$$
\mathcal{L}_t^{\text{Lm}} = \left\| \begin{pmatrix} M_{xx,t}^{00} & M_{xy,t}^{00} \\ M_{yx,t}^{00} & M_{yy,t}^{00} \end{pmatrix} - \begin{pmatrix} \Phi_{x,t} \\ \Phi_{y,t} \end{pmatrix} (\Psi_{x,t}, \Psi_{y,t}) \right\|_F^2,
\tag{I.2}
$$
$$
s.t. \quad \Phi_{y,t} = \Phi_{y,t-1}, \ \Psi_{y,t} = \Psi_{y,t-1},
$$

where $\Phi_{x,t}$ and $\Phi_{y,t}$ denote the sub-blocks of $\Phi_t$ for the dynamic and static landmarks, respectively. By adding such constraint of fixing static node representations, we have two advantages: (1) we can prevent updates to static node representations from bringing more noise and we can better utilize historical information to help improve model performance. (2) we can remove the smoothness item $\|C_t^i - C_{t-1}^i\|_F^2$ in Eq.(16), which can significantly reduce computation cost and further accelerate the our method.

The intra-information $\mathcal{L}_{i,t}^{\text{intra}}$ in Eq.(16) can be formulated as:

$$
\left\| \begin{pmatrix} M_{xx,t}^{ii} & M_{xy,t}^{ii} \\ M_{yx,t}^{ii} & M_{yy,t}^{ii} \end{pmatrix} - \begin{pmatrix} C_{x,t}^i \\ C_{y,t}^i \end{pmatrix} (H_{x,t}^i, H_{y,t}^i) \right\|_F^2 + \left\| \begin{pmatrix} M_{xx,t}^{0i} & M_{xy,t}^{0i} \\ M_{yx,t}^{0i} & M_{yy,t}^{0i} \end{pmatrix} - \begin{pmatrix} \Phi_{x,t} \\ \Phi_{y,t} \end{pmatrix} (H_{x,t}^i, H_{y,t}^i) \right\|_F^2 + \left\| \begin{pmatrix} M_{xx,t}^{i0} & M_{xy,t}^{i0} \\ M_{yx,t}^{i0} & M_{yy,t}^{i0} \end{pmatrix} - \begin{pmatrix} C_{x,t}^i \\ C_{y,t}^i \end{pmatrix} (\Psi_{x,t}, \Psi_{y,t}) \right\|_F^2 .
$$
$$
s.t. \ \Phi_{y,t} = \Phi_{y,t-1}, H_{y,t}^i = H_{y,t-1}^i, C_{y,t}^i = C_{y,t-1}^i, \Psi_{y,t} = \Psi_{y,t-1}.
\tag{I.3}
$$

The inter-information $\mathcal{L}_{i,t}^{\text{inter}}$ in Eq.(16) can be formulated as:

$$\left\| \begin{pmatrix} M_{xx,t}^{ij} & M_{xy,t}^{ij} \\ M_{yx,t}^{ij} & M_{yy,t}^{ij} \end{pmatrix} - \begin{pmatrix} C_{x,t}^{i} \\ C_{y,t}^{i} \end{pmatrix} (H_{x,t}^{j}, H_{y,t}^{j}) \right\|_{\text{F}}^{2} + \left\| \begin{pmatrix} M_{xx,t}^{ji} & M_{xy,t}^{ji} \\ M_{yx,t}^{ji} & M_{yy,t}^{ji} \end{pmatrix} - \begin{pmatrix} C_{x,t}^{j} \\ C_{y,t}^{j} \end{pmatrix} (H_{x,t}^{i}, H_{y,t}^{i}) \right\|_{\text{F}}^{2}.$$

$$s.t. \ C_{y,t}^{i} = C_{y,t-1}^{i}, H_{y,t}^{j} = H_{y,t-1}^{j}, C_{y,t}^{j} = C_{y,t-1}^{j}, H_{y,t}^{i} = H_{y,t-1}^{i}. \tag{I.4}$$

Interestingly, we can separate the bi-clustering item $\mathcal{L}_{i,t}^{\text{Bcr}} = Tr((F_t^i)^T L_{S_t^i} F_t^i)$ in Eq.(16) based on dynamic and static nodes. Combining Eqs.(I.2,I.3,I.4), the overall objective function of our proposed method can be modeled as:

$$O_t = \mathcal{L}_t^{\text{inter}} + \mathcal{L}_t^{\text{intra}} + \beta \mathcal{L}_t^{\text{Bcr}}. \tag{I.5}$$

Compared with Eq.(13), our method has two specific advantages. First, it eliminates the term $\|C_t^i - C_{t-1}^i\|_{\text{F}}^2$ for temporal smoothing, simplifying the optimization process. Second, it transforms the separability of temporal networks by exploiting only dynamic nodes, significantly reducing the time complexity.

Since $\mathcal{L}_t^{\text{Lm}}$ of Eq.(I.2) is optimized first to help enhance the consistency of representations between different subsets, we can fix $\Phi_t$ and $\Psi_t$ unchanged during the optimization of Eq.(16). First, we update the variable $\Phi_{x,t}$. We use $\mathcal{F}_{\Phi_{x,t}}$ to denote Eq.(I.2) as:

$$\mathcal{F}_{\Phi_{x,t}} = \left\| \begin{pmatrix} M_{xx,t}^{00} & M_{xy,t}^{00} \\ M_{yx,t}^{00} & M_{yy,t}^{00} \end{pmatrix} - \begin{pmatrix} \Phi_{x,t} \\ \Phi_{y,t} \end{pmatrix} (\Psi_{x,t}, \Psi_{y,t}) \right\|, \tag{I.6}$$

$$s.t. \ \Phi_{y,t} = \Phi_{y,t-1}, \Psi_{y,t} = \Psi_{y,t-1}.$$

By removing all irrelevant variables, Eq.(I.6) can be re-formulated as:

$$\begin{aligned} \mathcal{F}_{\Phi_{x,t}} =& \|M_{xx,t}^{00} - \Phi_{x,t}\Psi_{x,t}\|_{\text{F}}^2 + \|M_{xy,t}^{00} - \Phi_{x,t}\Psi_{y,t}\| \\ =& Tr((M_{xx,t}^{00} - \Phi_{x,t}\Psi_{x,t})^T (M_{xx,t}^{00} - \Phi_{x,t}\Psi_{x,t})) + Tr((M_{xy,t}^{00} - \Phi_{x,t}\Psi_{y,t})^T (M_{xy,t}^{00} - \Phi_{x,t}\Psi_{y,t})) \\ =& Tr((M_{xx,t}^{00})^T M_{xx,t}^{00} - (M_{xx,t}^{00})^T \Phi_{x,t}\Psi_{x,t} - (\Phi_{x,t}\Psi_{x,t})^T M_{xx,t}^{00} + (\Phi_{x,t}\Psi_{x,t})^T \Phi_{x,t}\Psi_{x,t} \\ & + (M_{xy,t}^{00})^T M_{xy,t}^{00} - (M_{xy,t}^{00})^T \Phi_{x,t}\Psi_{y,t} - (\Phi_{x,t}\Psi_{y,t})^T M_{xy,t}^{00} + (\Phi_{x,t}\Psi_{y,t})^T \Phi_{x,t}\Psi_{y,t}). \end{aligned} \tag{I.7}$$

We removed constant items $(M_{xx,t}^{00})^T M_{xx,t}^{00}$ and $(M_{xy,t}^{00})^T M_{xy,t}^{00}$, so that $\mathcal{F}_{\Phi_{x,t}}$ can be estimated by:

$$\mathcal{F}_{\Phi_{x,t}} = Tr(-(M_{xx,t}^{00})^T \Phi_{x,t}\Psi_{x,t} - (\Phi_{x,t}\Psi_{x,t})^T M_{xx,t}^{00} + (\Phi_{x,t}\Psi_{x,t})^T \Phi_{x,t}\Psi_{x,t} - (M_{xy,t}^{00})^T \Phi_{x,t}\Psi_{y,t} - (\Phi_{x,t}\Psi_{y,t})^T M_{xy,t}^{00} + (\Phi_{x,t}\Psi_{y,t})^T \Phi_{x,t}\Psi_{y,t}). \tag{I.8}$$

We can then obtain the gradient of $\Phi_{x,t}$ from Eq.(I.8) as:

$$\begin{aligned} \frac{\partial \mathcal{F}_{\Phi_{x,t}}}{\partial \Phi_{x,t}} =& \frac{\partial Tr(-(M_{xx,t}^{00})^T \Phi_{x,t}\Psi_{x,t})}{\partial \Phi_{x,t}} + \frac{\partial Tr(-(\Phi_{x,t}\Psi_{x,t})^T M_{xx,t}^{00})}{\partial \Phi_{x,t}} + \frac{\partial Tr((\Phi_{x,t}\Psi_{x,t})^T \Phi_{x,t}\Psi_{x,t})}{\partial \Phi_{x,t}} \\ & + \frac{\partial Tr(-(M_{xy,t}^{00})^T \Phi_{x,t}\Psi_{y,t})}{\partial \Phi_{x,t}} + \frac{\partial Tr(-(\Phi_{x,t}\Psi_{y,t})^T M_{xy,t}^{00})}{\partial \Phi_{x,t}} + \frac{\partial Tr((\Phi_{x,t}\Psi_{y,t})^T \Phi_{x,t}\Psi_{y,t})}{\partial \Phi_{x,t}} \\ =& -M_{xx,t}^{00}(\Psi_{x,t})^T - M_{xx,t}^{00}(\Psi_{x,t})^T + 2(\Phi_{x,t}\Psi_{x,t})(\Psi_{x,t})^T - M_{xy,t}^{00}(\Psi_{y,t})^T - M_{xy,t}^{00}(\Psi_{y,t})^T + 2(\Phi_{x,t}\Psi_{y,t})(\Psi_{y,t})^T \\ =& -2M_{xx,t}^{00}(\Psi_{x,t})^T - 2M_{xy,t}^{00}(\Psi_{y,t})^T + 2(\Phi_{x,t}\Psi_{x,t})(\Psi_{x,t})^T + 2(\Phi_{x,t}\Psi_{y,t})(\Psi_{y,t})^T. \end{aligned} \tag{I.9}$$

According to Lee et al. [25], we update $\Phi_{x,t}$ by using gradient descent as:

$$\Phi_{x,t} \leftarrow \Phi_{x,t} - \rho \frac{\partial \mathcal{F}_{\Phi_{x,t}}}{\partial \Phi_{x,t}}, \tag{I.10}$$

where $\rho$ is set to a small and positive number:

$$\rho = \frac{\Phi_{x,t}}{2(\Phi_{x,t}\Psi_{x,t})(\Psi_{x,t})^T + 2(\Phi_{x,t}\Psi_{y,t})(\Psi_{y,t})^T}. \tag{I.11}$$

Finally, our update rule for $\Phi_{x,t}$ is conducted as:

$$\Phi_{x,t} \leftarrow \Phi_{x,t} \cdot \frac{M_{xx,t}^{00}(\Psi_{x,t})^T + M_{xy,t}^{00}(\Psi_{y,t})^T}{(\Phi_{x,t}\Psi_{x,t})(\Psi_{x,t})^T + (\Phi_{x,t}\Psi_{y,t})(\Psi_{y,t})^T}. \tag{I.12}$$

Similarly, for $\Psi_{x,t}$, we have

$$\begin{aligned} \mathcal{F}_{\Psi_{x,t}} =& Tr(-(M_{xx,t}^{00})^T \Phi_{x,t}\Psi_{x,t} - (\Phi_{x,t}\Psi_{x,t})^T M_{xx,t}^{00} + (\Phi_{x,t}\Psi_{x,t})^T \Phi_{x,t}\Psi_{x,t} \\ & - (M_{yx,t}^{00})^T \Phi_{y,t}\Psi_{x,t} - (\Phi_{y,t}\Psi_{x,t})^T M_{yx,t}^{00} + (\Phi_{y,t}\Psi_{x,t})^T \Phi_{y,t}\Psi_{x,t}). \end{aligned} \tag{I.13}$$

The gradient for $\Psi_{x,t}$ is

$$\frac{\partial \mathcal{F}_{\Psi_{x,t}}}{\partial \Psi_{x,t}} = -2(\Phi_{x,t})^T M_{xx,t}^{00} - 2(\Phi_{y,t})^T M_{yx,t}^{00} + 2(\Phi_{x,t})^T \Phi_{x,t} \Psi_{x,t} + 2(\Phi_{y,t})^T \Phi_{y,t} \Psi_{x,t}.$$

The update rule for $\Psi_{x,t}$ is

$$\Psi_{x,t} \leftarrow \Psi_{x,t} - \rho \frac{\partial \mathcal{F}_{\Psi_{x,t}}}{\partial \Psi_{x,t}}, \tag{I.14}$$

where $\rho = \frac{\Psi_{x,t}}{2(\Phi_{x,t})^T \Phi_{x,t} \Psi_{x,t} + 2(\Phi_{y,t})^T \Phi_{y,t} \Psi_{x,t}}$.

Fixing $H_{y,t}^i$ and $C_{y,t}^i$, we can remove items without variables. $M_{yy,t}^{ii} \approx C_{y,t}^i H_{y,t}^i$, $M_{xy,t}^{0i} \approx \Phi_{x,t} H_{y,t}^i$ and $M_{yy,t}^{0i} \approx \Phi_{y,t} H_{y,t}^i$. The first item in Eq.(I.3) can be re-formulated as:

$$\|M_{xx,t}^{ii} - C_{x,t}^i H_{x,t}^i\|_F^2 + \|M_{xy,t}^{ii} - C_{x,t}^i H_{y,t}^i\|_F^2 + \|M_{yx,t}^{ii} - C_{y,t}^i H_{x,t}^i\|_F^2. \tag{I.15}$$

The second item of intra-information can be reduced as:

$$\|M_{xx,t}^{0i} - \Phi_{x,t} H_{x,t}^i\|_F^2 + \|M_{yx,t}^{0i} - \Phi_{y,t} H_{x,t}^i\|_F^2. \tag{I.16}$$

Let $M_{\cdot x,t}^{ij} = [M_{xx,t}^{ij}, M_{yx,t}^{ij}]^T$, Eq.(I.16) can be formulated as:

$$\|M_{\cdot x,t}^{0i} - \Phi_t H_{x,t}^i\|_F^2. \tag{I.17}$$

Similarly, by setting $M_{x\cdot,t}^{ij} = [M_{xx,t}^{ij}, M_{xy,t}^{ij}]$, we formulate the third item of intra-information for $M_t^{i0}$ in Eq.(I.3) as:

$$\|M_{x\cdot,t}^{i0} - C_{x,t}^i \Psi_t\|_F^2. \tag{I.18}$$

Combining Eqs.(I.15, I.17, I.18), intra-information loss of Eq.(I.3) can be reformulated as:

$$\mathcal{L}_{i,t}^{\text{intra}} = \underbrace{\|M_{\cdot x,t}^{0i} - \Phi_t H_{x,t}^i\|_F^2 + \|M_{x\cdot,t}^{i0} - C_{x,t}^i \Psi_t\|_F^2}_{\text{Landmark feature preservation}} + \underbrace{\|M_{xx,t}^{ii} - C_{x,t}^i H_{x,t}^i\|_F^2 + \|M_{xy,t}^{ii} - C_{x,t}^i H_{y,t}^i\|_F^2 + \|M_{yx,t}^{ii} - C_{y,t}^i H_{x,t}^i\|_F^2}_{\text{intra-information inside each subset}}. \tag{I.19}$$

Considering that $C_t^i, H_t^j$ can be represented as linear combinations of columns in $\Phi_t, \Psi_t$, we define $C_t^i = P_t^i \Phi_t$ and $H_t^j = \Psi_t Q_t^j$. Then, the first item of Eq.(I.4) is formulated as:

$$\left\| \begin{pmatrix} M_{xx,t}^{ij} & M_{xy,t}^{ij} \\ M_{yx,t}^{ij} & M_{yy,t}^{ij} \end{pmatrix} - \begin{pmatrix} P_{x,t}^i \\ P_{y,t}^i \end{pmatrix} \underbrace{\Phi_t (H_{x,t}^j, H_{y,t}^j)}_{M_t^{0j}} \right\|_F^2. \tag{I.20}$$

By removing the item related to $P_{y,t}^i$, Eq.(I.20) is formulated as:

$$\|M_{xx,t}^{ij} - P_{x,t}^i M_{x\cdot,t}^{0j}\|_F^2 + \|M_{xy,t}^{ij} - P_{x,t}^i M_{y\cdot,t}^{0j}\|_F^2 = \|M_{x\cdot,t}^{ij} - P_{x,t}^i M_t^{0j}\|_F^2. \tag{I.21}$$

Similarly, the second item of inter-information in Eq.(I.4) is

$$\|M_{xx,t}^{ji} - M_{x\cdot,t}^{j0} Q_{x,t}^i\|_F^2 + \|M_{yx,t}^{ji} - M_{y\cdot,t}^{j0} Q_{x,t}^i\|_F^2 = \|M_{\cdot x,t}^{ji} - M_t^{j0} Q_{x,t}^i\|_F^2. \tag{I.22}$$

Combining Eqs.(I.20, I.22), Eq.(I.4) can be re-formulated as:

$$\mathcal{L}_t^{\text{inter}} = \sum_{j \neq i} (\|M_{x\cdot,t}^{ij} - P_{x,t}^i M_t^{0j}\|_F^2 + \|M_{\cdot x,t}^{ji} - M_t^{j0} Q_{x,t}^i\|_F^2). \tag{I.23}$$

We can separate the bi-clustering item $\mathcal{L}_{i,t}^{\text{Bcr}}$ in Eq.(16) based on dynamic and static nodes. Specifically, we further deduce the following relation between clustering structure $S_{x,t}^i$ and landmark-related matrix $S_{P_{x,t}}^i$, as:

$$S_{x,t}^i = \begin{pmatrix} 0 & C_{x,t}^i \\ (C_{x,t}^i)^T & 0 \end{pmatrix} = \begin{pmatrix} I & 0 \\ 0 & \Phi_t^T \end{pmatrix} \underbrace{\begin{pmatrix} 0 & P_{x,t}^i \\ (P_{x,t}^i)^T & 0 \end{pmatrix}}_{S_{P_{x,t}}^i} \begin{pmatrix} I & 0 \\ 0 & \Phi_t \end{pmatrix}. \tag{I.24}$$

Since the left and right matrices of $S_{P_{x,t}}^i$ are scalars, the trace optimization of $C_{x,t}^i$ is proportional to the matrix $P_{x,t}^i$, i.e.,

$$Tr((F_t^i)^T L_{S_t^i} F_t^i) \propto Tr((F_{x,t}^i)^T L_{S_{x,t}^i} F_{x,t}^i) \propto Tr((F_{x,t}^i)^T L_{S_{P_{x,t}}^i} F_{x,t}^i), \tag{I.25}$$

Combining Eqs.(I.19,I.23,I.25), the overall objective function of our method can be modeled as

$$O_{i,t} = \mathcal{L}_{i,t}^{\text{intra}} + \mathcal{L}_{i,t}^{\text{inter}} + \beta \mathcal{L}_{i,t}^{\text{Bcr}}, \tag{I.26}$$

$$s.t. \ \mathcal{L}_{i,t}^{\text{Bcr}} = Tr((F_{x,t}^i)^T L_{S_{P_{x,t}^i}} F_{x,t}^i), \quad P_{x,t}^i \geq 0, \quad Q_{x,t}^i \geq 0, \quad P_{x,t}^i \mathbf{1} = \mathbf{1}, \quad (F_{x,t}^i)^T F_{x,t}^i = \text{I}.$$

By giving the converged $\Phi_t$ and $\Psi_t$, we follow Nie et al. [47] to iteratively update $F_{x,t}^i$, $Q_{x,t}^i$, and $P_{x,t}^i$ steps. Specifically, we first fix $P_{x,t}^i$ and $Q_{x,t}^i$ to update $F_{x,t}^i$ by transforming Eq.(I.27) into a trace optimization as:

$$\min_{F_{x,t}^i \in \mathcal{R}^{(|X_t^i|+r) \times k_t}, (F_{x,t}^i)^T F_{x,t}^i = I} Tr((F_{x,t}^i)^T L_{S_{P_{x,t}^i}} F_{x,t}^i), \tag{I.27}$$

where $|X_t^i|$ is the number of dynamic vertices, and the optimal $F_t^i$ is the $k_t$ smallest eigenvectors of $L_{S_{P_{x,t}^i}}$.

Analogously, we fix $F_{x,t}^i$ and $P_{x,t}^i$ to deduce Eq.(I.27) as:

$$\|M_{\cdot x,t}^{ii} - P_t^i M_t^{00} Q_{x,t}^i\| + \sum_{j \neq i} \|M_{\cdot x,t}^{ji} - M_t^{j0} Q_{x,t}^i\| + \|M_{\cdot x,t}^{0i} - M_t^{00} Q_{x,t}^i\|, \tag{I.28}$$

and obtain the update rule for $Q_{x,t}^i$ as:

$$Q_{x,t}^i \leftarrow Q_{x,t}^i \cdot \frac{(M_t^{00})^T J_1 + \sum_{j \neq i} (M_t^{j0})^T M_{\cdot x,t}^{ji}}{(M_t^{00})^T J_2 M_t^{00} Q_{x,t}^i + \sum_{j \neq i} (M_{\cdot x,t}^{j0})^T M_t^{j0} Q_{x,t}^i}, \tag{I.29}$$

where $J_1 = (P_t^i)^T M_{\cdot x,t}^{ii} + M_{\cdot x,t}^{0i}$, and $J_2 = I + (P_{x,t}^i)^T P_{x,t}^i + (P_{y,t}^i)^T P_{y,t}^i$.

When fixing $Q_{x,t}^i$ and $F_{x,t}^i$ to consider $P_{x,t}^i$ as the variable, Eq.(I.27) is deduced as:

$$\|M_{x \cdot,t}^{ii} - P_{x,t}^i M_t^{00} Q_t^i\| + \sum_{j \neq i} \|M_{x \cdot,t}^{ij} - P_{x,t}^i M_t^{0j}\| + \|M_{x \cdot,t}^{i0} - P_{x,t}^i M_t^{00}\| + \beta Tr((F_{x,t}^i)^T L_{S_{P_{x,t}^i}} F_{x,t}^i). \tag{I.30}$$

In accordance with $Tr(F^T L_S F) = \sum_{a=1}^n \sum_{b=1}^r \|\mathbf{f}_{a\cdot} - \mathbf{f}_{n+b\cdot}\|_2^2 c_{ab}$, the trace optimization in Eq.(I.30) can be formulated as:

$$Tr((F_{x,t}^i)^T L_{S_{P_{x,t}^i}} F_{x,t}^i) = \sum_{a=1}^{|X_t^i|} \sum_{b=1}^r \|\mathbf{f}_{a\cdot,xt}^i - \mathbf{f}_{|X_t^i|+b\cdot,xt}^i\|_2^2 \ p_{ab,xt}^i, \tag{I.31}$$

where the vector $\mathbf{f}_{a\cdot,xt}^i$ is the $a$-th row of $F_{x,t}^i$, and $p_{ab,xt}^i$ is the element at the $a$-th row and $b$-th column of $P_{x,t}^i$. By substituting Eq.(I.31) into Eq.(I.30), the objective function is re-formulated as:

$$O_{i,t}^{II,P} = \|M_{x \cdot,t}^{ii} - P_{x,t}^i M_t^{00} Q_t^i\| + \sum_{j \neq i} \|M_{x \cdot,t}^{ij} - P_{x,t}^i M_t^{0j}\| + \|M_{x \cdot,t}^{i0} - P_{x,t}^i M_t^{00}\| + \beta \sum_{a=1}^{|X_t^i|} \sum_{b=1}^r \|\mathbf{f}_{a\cdot,xt}^i - \mathbf{f}_{|X_t^i|+b\cdot,xt}^i\|_2^2 \ p_{ab,xt}^i. \tag{I.32}$$

Because Eq.(I.32) only involves $P_{x,t}^i$, by setting $\partial O_{i,t}^{II,P}/\partial \mathbf{p}_{a\cdot,xt}^i = 0$, we independently update each row $\mathbf{p}_{a\cdot,xt}^i$ of $P_{x,t}^i$ as:

$$\mathbf{p}_{a\cdot,xt}^i \leftarrow \mathbf{p}_{a\cdot,xt}^i \cdot \frac{(\boldsymbol{\gamma}_1 (Q_t^i)^T + \boldsymbol{\gamma}_2)(M_t^{00})^T + \sum_{j \neq i} \boldsymbol{\gamma}_3 (M_t^{0j})^T}{\mathbf{p}_{a\cdot,xt}^i M_t^{00} J_3 + \sum_{j \neq i} \mathbf{p}_{a\cdot,xt}^i M_t^{0j} (M_t^{0j})^T + \beta \mathbf{e}_{a\cdot,xt}^i}, \tag{I.33}$$

where $J_3 = Q_t^i (M_t^{00} Q_t^i)^T + (M_t^{00})^T$, $\mathbf{e}_{a\cdot,xt}^i$ is a vector, the $b$-th element of which is $e_{ab,xt}^i = \|\mathbf{f}_{a\cdot,xt}^i - \mathbf{f}_{|X_t^i|+b\cdot,xt}^i\|_2^2$. The $\boldsymbol{\gamma}_1$, $\boldsymbol{\gamma}_2$, and $\boldsymbol{\gamma}_3$ are the $a$-th row of $M_{x \cdot,t}^{ii}$, $M_{x \cdot,t}^{i0}$, and $M_{x \cdot,t}^{ij}$, respectively.

Received 20 February 2007; revised 12 March 2009; accepted 5 June 2009

