# OpenReview forum: "Revisiting Dynamic Graph Clustering via Matrix Factorization"
_ACM.org/TheWebConf/2025/Conference — WWW 2025 Oral_

### Official Review · Reviewer_56Cn · 2024-11-28

**Novelty:** 3
**Technical Quality:** 5

**Review:**

In this work, the authors give a new technique for dynamic/temporal graph clustering using a novel twist on matrix factorization. Dynamic graph clustering is a well-studied problem in practice, and has been done via neural network-based approaches and other matrix factorization-based approaches in a long sequence of prior work. This prior work has two key shortcomings: 1. MF-based algorithms tend to be slow, as the basic MF problem is NP-hard. 2. Neural net based methods are faster, but do not obtain as high quality of clusterings due to the fact that they primarily learn embeddings.

To overcome these shortcomings, the authors propose a method DyG-MF that constructs a sequence of matrix factorizations that are consistent with each other. To improve scalability, the authors use a K-means-like paradigm where temporal landmarks are selected (like cluster centers) and are used to cluster the rows of an adjacency-related matrix called the PMI matrix. This clustering is then used to construct the factorization. The temporal landmarks are chosen by solving an optimization problem, where the objective function of that optimization problem involves several different terms: the basic factorization-error term, a consistency term connecting rows from adjacent time steps, and a bi-clustering regularization term to handle noise. There are many hyper parameters present in the algorithm definition, like the weighting of the different objective terms and the number of temporal landmarks chosen in each step.

The authors have theorems in their paper, but they do not contribute to the evaluation method and they only serve to validate basic properties of their algorithm. They do have a detailed experimental evaluation of DyG-MF, done on a variety of synthetic and real-world benchmarks. The synthetic benchmarks model the discovery of a pre-chosen underlying sequence of cluster formation and dissolution events, while clusters in the real-world graphs are less clear and are compared against previous methods.

The results are encouraging and DyG-MF does look more scalable than previous methods. However, I would appreciate a clearer conceptual message in the paper. The authors introduce a variety of techniques, like bi-clustering regularization and selective embedding update, where the value is somewhat unclear. In particular, I am somewhat concerned that these techniques were included specifically to boost performance enough to beat previous benchmarks. The authors did give an ablation study illustrating the utility of these algorithmic components, but I would appreciate a stronger conceptual description of why these techniques must be used. It is also unclear how important factors like hyperparameter tuning were to the results, as while DyG-MF does outperform other methods, it does not do so by a large margin. Thus, DyG-MF looks like a conceptually incremental improvement on top of a long line of work on dynamic graph clustering, so I recommend weak rejection.

**Questions:**

Key Questions:

Is there a key selling point of DyG-MF over previous work? At the end of section 1, several minor selling points are given — better scalability, better robustness to noise, and better consistency over time — but the message would be stronger if the authors could really emphasize one metric on which this algorithm is much better than previous work. It is clear from the results that DyG-MF at least matches all previous methods on all metrics, but is there one metric that it performs especially well in?

In the theorem statements, particularly Theorem 1 and Theorem 2, there are several concepts that are not formally defined like “ensuring embedding consistency between subsets of nodes” and “The bi-clustering regularization.” How exactly are these defined?

Other minor questions and comments that would be nice to have responses to:

Line 95: “Matrix factorization is an NP-hard problem with time complexity of approximately…” Polynomial time in spite of being an NP-hard problem? This doesn’t make sense — emphasize that NP-hardness is for some versions of matrix factorization

Line 267: Shouldn’t the objective be \sum_t O_t rather than just one O_t?

Line 281: Why is it just n^3? Shouldn’t it be NP-hard to optimize this?

Line 370: What does this theorem mean? “Ensuring embedding consistency between subsets of nodes” has not been defined.

Line 415, Theorem 3 statement: The bi-clustering regularization has not yet been formally defined; it has only been alluded to.

Line 539: What do dynamic communities mean in the real-world graphs? What do they look like conceptually?

Line 581, Table 2: Should it be clear that most neural net-based methods are worse than MF-based methods from this table? The introduction made it clear that neural net-based methods should be worse as they can only optimize embeddings, not the clustering end product. It would be nice to add a discussion about this phenomenon.

Line 599, Table 2: What does 100 mean here? Is this perfect performance? Why should we expect DyG-MF to do perfectly? Also what does the p-value mean here? How are confidence intervals determined?

Line 916: “In this study, we proposed a novel scalable and robust temporal separated matrix factorization method to reveal the evolution mechanism of complex real-world complex systems.” This sentence is quite vague — is it possible to be more specific?

**Reviewer Confidence:**

3: The reviewer is confident but not certain that the evaluation is correct

**Scope:**

4: The work is relevant to the Web and to the track, and is of broad interest to the community

---

### Official Review · Reviewer_jDbX · 2024-11-29

**Novelty:** 5
**Technical Quality:** 5

**Review:**

This paper introduces DyG-MF, a dynamic graph clustering framework based on matrix factorization. It tackles scalability, robustness, and efficiency issues in clustering large-scale, noisy, real-world dynamic graphs. Experiments on synthetic and real-world dynamic graphs demonstrate that DyG-MF outperforms state-of-the-art methods in clustering quality, scalability, and robustness.

Pros
1. DyG-MF integrates three novel strategies—Temporal Separated Matrix Factorization (TSMF), Bi-clustering Regularization (BR), and Selective Embedding Updating (SEU)—that collectively enhance clustering performance.
2. Experiments highlight that DyG-MF scales to massive graphs with over 3 million nodes while maintaining high robustness and efficiency, outperforming existing methods.
3. The paper is well written and easy to follow.

Cons
1. While the authors focus on reducing time complexity, the reported space complexity, $O(|V_t|^2)$, may pose challenges for graphs with millions of nodes. A comparison with the space complexity of other methods would provide better context.
2. The paper does not specify the hardware used for experiments. Providing details about the computational setup (e.g., GPU/CPU specs, memory) would improve reproducibility and help assess the practicality of DyG-MF on large datasets.
3. The reduction in time complexity might be achieved at the expense of increased memory usage. A deeper analysis of this trade-off is missing.

**Questions:**

Please see the Cons above.

**Reviewer Confidence:**

2: The reviewer is willing to defend the evaluation, but it is likely that the reviewer did not understand parts of the paper

**Scope:**

4: The work is relevant to the Web and to the track, and is of broad interest to the community

---

### Official Review · Reviewer_Q3Mz · 2024-12-01

**Novelty:** 5
**Technical Quality:** 5

**Review:**

## Summary
This paper focuses on the problem of dynamic graph clustering, which aims to identify time-varying clusters in dynamic graphs. Specifically, this work is inspired by matrix factorization-based methods and seeks to overcome their scalability and robustness challenges. To tackle the scalability issue, the authors propose a temporal-separated matrix factorization approach, which divides the original large-scale matrix into smaller sub-matrices for independent factorization. In this process, a temporal landmark selection strategy is introduced to ensure consistent and coherent node embeddings across all sub-matrix factorizations. To enhance robustness, the authors introduce a bi-clustering regularization term to filter out noisy features from graph embeddings. Additionally, a selective embedding update strategy is employed to improve model effectiveness and efficiency by dividing nodes into static and dynamic groups, updating only the embeddings of nodes in the dynamic group.
## Strengthes
1. The motivation is clear, which stems from the scalability and robustness issues of existing matrix factorization-based methods. In addition, a pre-experiment and a case study are provided in Figure 1.
2. The proposed method addresses the issues of existing dynamic graph clustering methods from multiple perspectives, and demonstrates superior performance, scalability and robustness in experiments.
3. The article is well-structured and easy to follow.
## Weaknesses
1. Given that the core of this work is graph node clustering, clustering algorithms are of crucial importance. This paper is primarily based on the K-means clustering algorithm; however, there are many other clustering algorithms, which are not discussed or compared in this paper.
2. Although dividing dynamic nodes and static nodes based on the degree of topological changes and position drift seems reasonable, the effectiveness of the two terms in Eq.14, namely "topological changes" and "relative positions shift," still lacks theoretical evidence. In particular, for "relative positions shift," if the nearest cluster center to the same node varies significantly at different time points, the value of this term seems to be uncertain and unstable.

**Questions:**

1. How would the performance of the proposed method change if the node clustering algorithm were replaced with a different clustering algorithm instead of K-means?
2. Apart from the ablation experiment in Section 5.4, is it possible to theoretically prove the rationale and effectiveness of the two terms in Eq. 14 separately?

**Reviewer Confidence:**

3: The reviewer is confident but not certain that the evaluation is correct

**Scope:**

3: The work is somewhat relevant to the Web and to the track, and is of narrow interest to a sub-community

---

### Official Review · Reviewer_43zU · 2024-12-02

**Novelty:** 5
**Technical Quality:** 4

**Review:**

**Summary:**
The paper presents a novel approach to dynamic graph clustering, focusing on scalability, robustness, and effectiveness. The authors propose three main contributions: temporal separated matrix factorization to improve scalability, bi-clustering regularization to enhance robustness, and selective embedding updating for efficiency. Experimental results on synthetic and real-world datasets show the advantages of the proposed method, demonstrating its ability to address some common challenges in dynamic graph clustering.

**Strengths:**
1. **Novelty of Contributions**: The proposed temporal separated matrix factorization and bi-clustering regularization are valuable contributions. The idea of improving scalability by dividing the matrix into smaller parts and optimizing graph embedding and clustering simultaneously is well thought out. These ideas address key challenges in dynamic graph clustering, particularly with large-scale and noisy data.

2. **Experimental Validation**: The experimental section provides a comprehensive evaluation across several synthetic and real-world datasets, highlighting the proposed method’s advantages in terms of scalability, robustness, and effectiveness. The comparison with existing methods strengthens the claims made by the authors.

3. **Clear and Structured Writing**: The paper is well-organized and clearly written, with a solid explanation of the problem, approach, and experimental setup. The results are presented in a manner that is easy to follow.

**Weaknesses:**
1. **Scalability and Real-World Application**: While the authors claim scalability improvements, the experiments do not provide enough insight into the method's performance on very large, real-world dynamic graphs, especially those with millions of nodes and edges. Including benchmarks with larger, more complex datasets would further validate the scalability claims.

2. **Insufficient Discussion on Limitations**: The paper could benefit from a more detailed discussion of the limitations of the proposed method. For example, how does the method perform under extreme data sparsity or with highly imbalanced clusters? Are there any potential trade-offs between the improved scalability and clustering quality?

**Questions:**

See in weakness.

**Reviewer Confidence:**

2: The reviewer is willing to defend the evaluation, but it is likely that the reviewer did not understand parts of the paper

**Scope:**

3: The work is somewhat relevant to the Web and to the track, and is of narrow interest to a sub-community

---

### Official Review · Reviewer_2Pin · 2024-12-03

**Novelty:** 6
**Technical Quality:** 6

**Review:**

1. The paper introduces three innovative techniques: temporal separated matrix factorisation, bi-clustering regularisation, and selective embedding updating, which are well-motivated to address scalability and robustness challenges in dynamic graph clustering.

2. The experimental results span synthetic and real-world dynamic graphs, clearly demonstrating scalability, robustness, and effectiveness compared to the state-of-the-art baselines, which is impressive.

3. The proposed method, DyG-MF, significantly improves scalability by reducing the computational complexity from O(n3) to O(n2), which is also validated in the experiments conducted.

4. Bi-clustering regularisation effectively filters out noisy data, maintaining high clustering performance even under small noise perturbations.

5. The methodology is well-structured, with clear mathematical formulations, insightful theoretical proofs and practical illustrations of its concepts. The code was also provided for better validation and understanding.

**Questions:**

1. The time complexity analysis omits a detailed comparison with neural network-based methods, which might provide additional insights into practical trade-offs. Also, the time complexities are dependent on the number of iterations. A more detailed analysis could have been provided with some experiments on how the final equations rely on the number of iterations in terms of accuracy and time.

2. Although some ablation studies are conducted, the individual contributions of each module (e.g., landmark selection vs. bi-clustering regularisation) could have been more exhaustively evaluated. This could have been done by more rigorous experiments on how the overall performance depends on each module.

3. While the authors mention code availability, further details on datasets and preprocessing steps for replicability are sparse in the main text, leaving room for improvement in reproducibility.

4. In line 94, the authors mention that matrix factorisation is NP-hard but also describe its complexity as O(n3). This inconsistency raises questions about whether the scalability claims are rigorously substantiated and require further clarification.

5. As defined in Eq. (3), the temporal smoothing component prioritises the global smoothness of node embeddings across consecutive timestamps but lacks fine-grained handling of nodes with highly dynamic behaviours. This can result in over-penalizing significant temporal changes in nodes, potentially degrading performance in datasets with abrupt community shifts or irregular evolutionary patterns. A more refined approach to distinguish between stable and highly dynamic nodes could improve this aspect.

6. A not-so-important feedback is to check the spelling mistakes, like seperated (line 108), optomize (Figure 2e), subsects (Figure 2b), graphsm (line 280).

Feedback

The paper presents an innovative framework, DyG-MF, for dynamic graph clustering, introducing temporal separated matrix factorisation, bi-clustering regularisation, and selective embedding updating to address scalability, robustness, and efficiency challenges. The methodology is well-written and supported by extensive experiments on synthetic and real-world datasets, demonstrating significant improvements over 14 state-of-the-art baselines in performance metrics like NMI, NF1, and modularity. However, certain aspects could be improved (see above points). The experiments consider only a limited number of snapshots (5–10), which constrains the insights into the method's behaviour over long-term graph evolution. Despite these limitations, the proposed contributions are significant and represent a good and novel step forward in dynamic graph clustering.

**Reviewer Confidence:**

3: The reviewer is confident but not certain that the evaluation is correct

**Scope:**

4: The work is relevant to the Web and to the track, and is of broad interest to the community